# Mechanisms underlying neonate-specific metabolic effects of volatile anesthetics

Julia Stokes[1†], Arielle Freed[1,2†], Rebecca Bornstein[3†], Kevin N Su[4], John Snell[1], Amanda Pan[1], Grace X Sun[1], Kyung Yeon Park[1], Sangwook Jung[1], Hailey Worstman[1], Brittany M Johnson[1], Philip G Morgan[1,4], Margaret M Sedensky[1,4], Simon C Johnson[1,3,4,5]*

[1]Center for Integrative Brain Research, Seattle Children's Research Institute, Seattle, United States; [2]University of Washington School of Dentistry, Seattle, United States; [3]Department of Pathology, University of Washington, Seattle, United States; [4]Department of Anesthesiology and Pain Medicine, University of Washington, Seattle, United States; [5]Department of Neurology, University of Washington, Seattle, United States

**Abstract** Volatile anesthetics (VAs) are widely used in medicine, but the mechanisms underlying their effects remain ill-defined. Though routine anesthesia is safe in healthy individuals, instances of sensitivity are well documented, and there has been significant concern regarding the impact of VAs on neonatal brain development. Evidence indicates that VAs have multiple targets, with anesthetic and non-anesthetic effects mediated by neuroreceptors, ion channels, and the mitochondrial electron transport chain. Here, we characterize an unexpected metabolic effect of VAs in neonatal mice. Neonatal blood β-hydroxybutarate (β-HB) is rapidly depleted by VAs at concentrations well below those necessary for anesthesia. β-HB in adults, including animals in dietary ketosis, is unaffected. Depletion of β-HB is mediated by citrate accumulation, malonyl-CoA production by acetyl-CoA carboxylase, and inhibition of fatty acid oxidation. Adults show similar significant changes to citrate and malonyl-CoA, but are insensitive to malonyl-CoA, displaying reduced metabolic flexibility compared to younger animals.

*For correspondence:
simoncj@u.washington.edu

[†]These authors contributed equally to this work

## Introduction

Volatile anesthetic agents (VAs) have been routinely used for general anesthesia for over 150 years; their development represented a major advance in human medicine (*Whalen et al., 2005*). Despite their prevalence, the precise targets of VAs, and mechanisms underlying their pleiotropic effects, are largely undefined. While most intravenous anesthetics appear to work through one or a small number of functional targets, such as neuroreceptors, VAs have been shown to interact with and impact a wide range of molecules and physiologic functions. Competing hypotheses currently exist to explain the precise anesthetic mechanisms of VAs, but general disruption of membrane bound proteins, either selectively or en masse, is a common feature among favored models (*Weinrich and Worcester, 2018*; *Herold et al., 2017*; *Sidebotham and Schug, 1997*).

In addition to their desired neurologic effects (e.g. analgesia, paralysis, amnesia, and sedation), VAs have a range of both beneficial and detrimental off-target effects in various organ systems, including immune modulation, tumor enhancement, and cardioprotection (*Stollings et al., 2016*; *Sekandarzad et al., 2017*; *Lorsomradee et al., 2008*). As in the case of anesthesia, the mechanisms underlying VA effects in non-neuronal tissues are enigmatic more often than not. Defining the mechanisms of VA action in a given setting is complicated by the diverse physiologic and molecular effects of VAs – it has been remarkably difficult to isolate and define individual mechanistic pathways involved in the effects of VAs. Experimental approaches to studying VAs are hampered by their

weak interactions with targets and the limitations of volatile (gaseous), poorly water soluble, agents, which together preclude many of the tools used to study intravenous anesthetic agents.

Routine anesthesia with VAs is considered to be safe in healthy individuals, but anesthetic sensitivity and toxicity have been demonstrated in certain clinical populations defined by either age or genetic makeup. In many cases, the precise underpinnings of hypersensitivity remain poorly understood. Known sensitive populations include those with genetic defects in mitochondrial electron transport chain complex I (ETC CI), which lead to profound hypersensitivity to VAs, or individuals with mutations in the ryanodine receptor RYR1, who can experience malignant hyperthermia upon exposure to VAs (*Niezgoda and Morgan, 2013*; *Rosenberg et al., 2015*). Additionally, in recent years, there has been a recognition that neonatal mammals, and developing invertebrate animals, are sensitive to CNS damage as a result of extended or repeat exposure to VAs; this concept of potential anesthetic induced neurotoxicity represented a paradigm shift in pediatric anesthesia (*Johnson et al., 2019a*; *Na et al., 2017*). While the clinical relevance of paradigms used to study these phenomena are an area of active debate, and many distinct mechanisms have been proposed to mediate these toxic effects of VAs, it is clear that VA exposure can induce CNS injury under certain conditions (*Johnson et al., 2019a*; *McCann and Soriano, 2019*; *Johnson et al., 2019b*). Mechanistic studies defining the differential effects of VAs on neonates versus older animals have not been available.

Here, we identify a surprising and previously undocumented metabolic effect of VAs specific to neonatal animal. Our data reveal both the mechanism of this effect and the nature of the difference between the neonatal and adolescent mice in response to VA exposure.

## Results

### Metabolic status of neonatal mice is rapidly disrupted by volatile anesthesia exposure

Neonatal mice (post-natal day 7, P7) are in a ketotic state compared to adolescent (post-natal day 30, PD30) or young adult (P60) animals, as has been previously reported (*Figure 1*; *Johnson et al., 2019b*; *Bougneres et al., 1986*; *Cotter et al., 2011*; *Williamson, 1985*; *Edmond et al., 1985*). Steady-state blood β-hydroxybutyrate (β-HB) is ~2 mM in P7 pups, below 1 mM in P30 mice, and approximately 0.5 mM in P60 animals. In contrast, neonatal animals have low average resting glucose relative to adolescent and adult animals, with an average glucose of ~160 mg/dL at P7 compared to ~230 and ~260 mg/dL in P30 and P60 animals, respectively.

We recently reported that exposure of P7 neonatal mice to an anesthetizing dose of 1.5% isoflurane leads to a significant reduction in circulating levels of the ketone β-HB by 2 hr of exposure (*Johnson et al., 2019b*), a physiologic effect of anesthesia that had not been previously reported. To further investigate this phenomenon, we assessed β-HB levels as a function of exposure time in P7 mice (see Materials and methods). Exposure to isoflurane resulted in an extremely rapid reduction in circulating β-HB, with an effect half-life of less than 12 min and a significant reduction compared to baseline by 7.5 min of exposure (*Figure 1C*). After reaching a valley of ~1 mM by 30 min of exposure, β-HB remained low to 2 hr. Littermate neonates removed from their parents and placed in control conditions (conditions matching anesthesia exposed, but in ambient air, see Materials and methods) show a slight decrease in β-HB at 30 min followed by a time-dependent increase in blood β-HB up to 2 hr. Pairwise comparisons of isoflurane- and control-exposed animals demonstrates highly significant reductions in β-HB in the isoflurane-exposed group at each timepoint (*Figure 1D*). 1.5% isoflurane also led to a significant increase in lactate by 60 min (see *Figure 1—figure supplement 1*; see also ETC CI).

While isoflurane rapidly depleted circulating ketones in neonates, isoflurane anesthesia had no impact on this circulating ketone in older (P30) animals (*Figure 1E,F*). Moreover, both control (fasted) and isoflurane-exposed P30 mice show a slight but statistically significant *increase* in β-HB by 2 hr (*Figure 1E*).

P7 neonatal mice exposed to 1.5% isoflurane anesthesia fail to maintain normal blood glucose homeostasis (*Figure 1G–H*; *Johnson et al., 2019b*). Following an initial increase in glucose in the 1.5% isoflurane-exposed pups, blood glucose falls more rapidly in isoflurane-exposed animals than in controls; both groups are significantly lower than baseline by 60 min and continue to fall

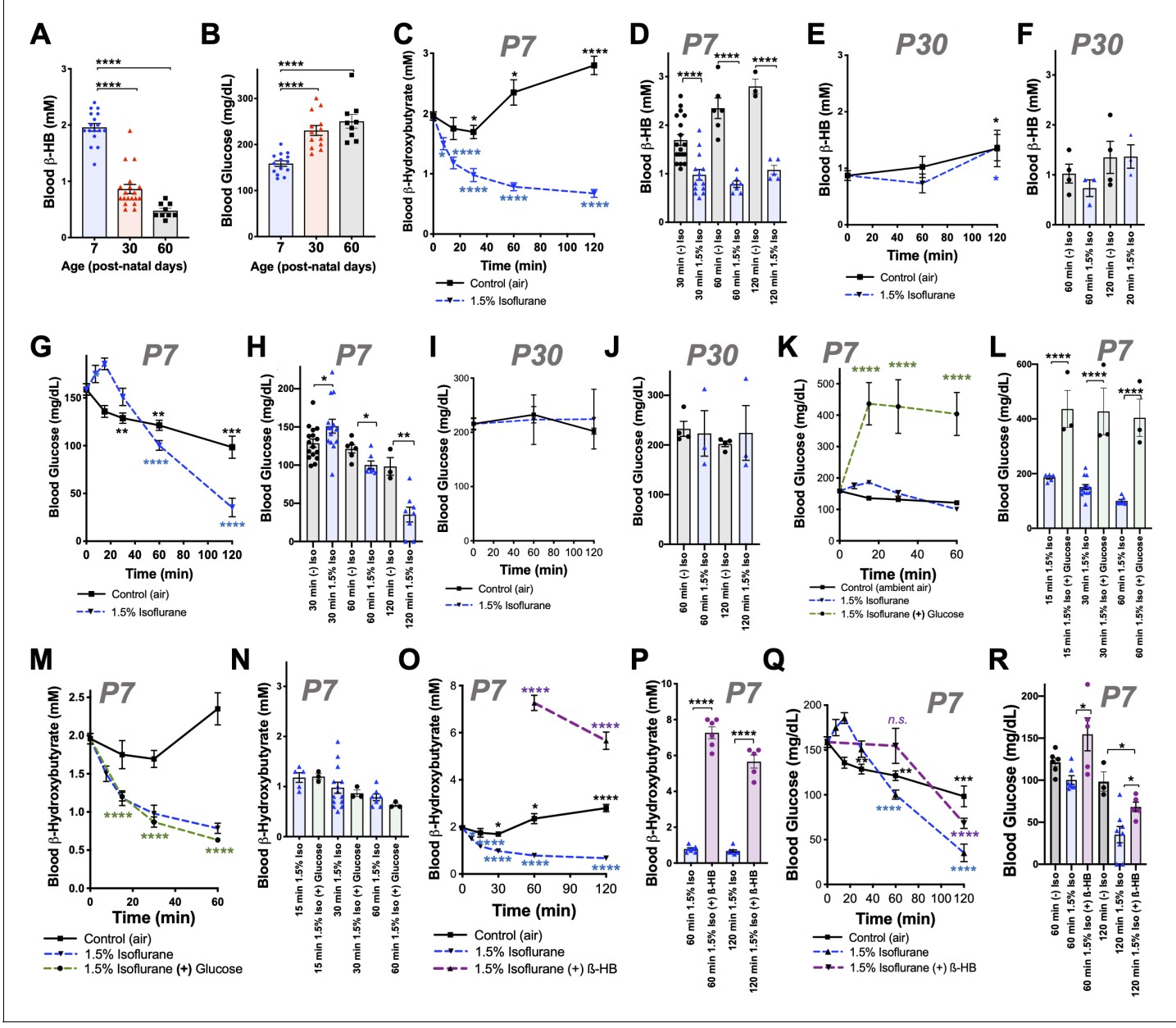

**Figure 1.** Isoflurane exposure disrupts circulating glucose and beta-hydroxybutyrate in neonatal mice. (**A**) Blood β-hydroxybutyrate (β-HB) concentration in neonatal post-natal day 7 (P7), adolescent post-natal day 30 (P30), and young adult post-natal day 60 (P60) mice. n = 17, 19, and 9, respectively. ****p<0.0001 by pairwise t-test. ANOVA ****p<0.0001. (**B**) Blood glucose in P7, P30, and P60 mice. n = 14, 13, and 9, respectively. ****p<0.0001 by pairwise t-test. ANOVA ****p<0.0001. (**C, D**) Blood β-HB in P7 neonatal mice exposed to 1.5% isoflurane or control conditions for 0 to 120 min. (**C**) Pairwise comparisons shown between baseline (time = 0) and respective treatment timepoints. *p≤0.05, ****p<0.0001 by two-tailed pairwise t-test. (**D**) Bar graphs with individual datapoints for pairwise comparisons between treatments at 30, 60, and 120 min. ****p<0.0001 by two-tailed pairwise t-test. (**E, F**) Blood β-HB in P30 adolescent mice exposed to 1.5% isoflurane or control conditions for 0–120 min. (**E**) Control and isoflurane-exposed groups both show a modest but significant increase in β-HB over baseline by 2 hr of exposure, *p≤0.05. (**F**) Isoflurane exposure did not significantly alter β-HB levels relative to time-matched control conditions. (**G–H**) Blood β-HB in P7 neonatal mice exposed to 1.5% isoflurane or control conditions for 0–120 min. (**G**) Pairwise comparisons shown between baseline (time = 0) and respective treatment timepoints. *p≤0.05, **p<0.005, ***p<0.0005, ****p<0.0001 by two-tailed pairwise t-test. (**H**) Bar graphs with individual datapoints for pairwise comparisons between treatments at 30, 60, and 120 min. *p≤0.05, **p<0.005. (**I, J**) Blood glucose levels in P30 mice exposed to control conditions or 1.5% isoflurane anesthesia for up to 120 min. (**I**) Blood glucose levels did not significantly change compared to baseline. (**J**) Bar graphs of data in (**I**) with individual datapoints for pairwise comparison of blood glucose levels by treatment. No significant changes observed. (**K, L**) Blood glucose in P7 mice provided glucose by IP injection at the start of anesthetic exposure plotted as a function of time. (**K**) Control-exposed and 1.5% isoflurane- exposed data from (**G**) shown for reference. Pairwise comparisons shown between baseline (T = 0) values and 15, 30, and 60 min timepoints in 1.5% isoflurane (+) glucose treatment group, ****p<0.0001 by pairwise

*Figure 1 continued on next page*

*Figure 1 continued*

t-test. (L) Bar graphs of (K) with individual datapoints for pairwise comparisons of blood glucose in mice exposed to 1.5% isoflurane or 1.5% isoflurane (+) glucose. ****p<0.0001 by two-tailed pairwise t-test. (M, N) Blood β-HB levels in mice provided glucose by IP injection at the start of anesthetic exposure, plotted as a function of time. (M) Control-exposed and 1.5% isoflurane-exposed data from (C) shown for reference. Pairwise comparisons shown between baseline (T = 0) and 15, 30, and 60 min timepoints in 1.5% isoflurane (+) glucose treatment group, ****p<0.0001. (N) Bar graphs of (M) with individual datapoints for pairwise comparisons of blood β-HB in mice exposed to 1.5% isoflurane or 1.5% isoflurane (+) glucose. Glucose administration did not attenuate the loss of β-HB in response to isoflurane exposure. (O, P) Blood β-HB levels in mice provided β-HB by IP injection at the start of anesthetic exposure. (O) Control-exposed and 1.5% isoflurane-exposed data from (C) shown for reference. Pairwise comparisons shown between baseline (T = 0) and 60 and 120 min timepoints in 1.5% isoflurane (+) β-HB treatment group, ****p<0.0001. (P) Bar graphs of (O) with individual datapoints for pairwise comparisons of blood β-HB in mice exposed to 1.5% isoflurane or 1.5% isoflurane (+) glucose. ****p<0.0001. (Q, R) Blood glucose levels in mice provided β-HB by IP injection at the start of anesthetic exposure, plotted as a function of time. (Q) Control -exposed and 1.5% isoflurane-exposed data from (G) shown for reference. Pairwise comparisons shown between baseline (T = 0) and 60 and 120 min timepoints in 1.5% isoflurane (+) β-HB treatment group, ****p<0.0001, n.s. – not significant. (R) Bar graphs of (Q) with individual datapoints for pairwise comparisons of blood β-HB in mice exposed to 1.5% isoflurane or 1.5% isoflurane (+) glucose or baseline. *p<0.05. For all data, error bars represent standard error of the mean (SEM). ANOVA p-value for 1 hr dataset *p=0.0083; ANOVA for 2 hr dataset **p=0.0033. (A–R) For all data, n ≥ 3 per time/treatment. Each datapoint in bar graphs represents an individual animal. See Materials and methods for additional details.

The online version of this article includes the following figure supplement(s) for figure 1:

**Figure supplement 1.** Dose-dependent effects of 1% and 1.5% isoflurane on blood lactate and glucose in P7 mice.

**Figure supplement 2.** Circulating nutrient factors.

**Figure supplement 3.** Oxygen concentration does not impact circulating glucose or β-HB.

(*Figure 1G*). Blood glucose levels are lower in the isoflurane group than fasted (control treated) pups at 60 and 120 min (*Figure 1H*), falling to ~35 mg/dL by 2 hr of isoflurane exposure, substantially below the ~100 mg/dL observed in controls. In contrast to neonates, isoflurane anesthesia has no impact on circulating glucose levels in 30 day old animals exposed for up to 2 hr; glucose is also maintained over this time in control-exposed animals (*Figure 1I–J*).

We considered the possibility that VA-induced changes to circulating metabolic hormones may underlie the changes observed, but analysis of key factors (see Materials and methods) ruled out this possibility. Insulin, glucagon, and related factors were significantly altered by 30 min of VA exposure, but the directionality of changes was inconsistent with driving the observed changes (see *Figure 1— figure supplement 2* for details). Furthermore, both 1.5% and 1% isoflurane induced similar changes by 30 min, while metabolic effects on glucose and lactate are differential between these doses (see below).

We also considered the potential impact of oxygen levels in these exposures. In accordance with standard practice in pre-clinical rodent studies, VAs were delivered using 100% oxygen as a carrier gas. To test whether oxygen exposure acutely impacted β-HB or glucose levels, we exposed P7 neonates to 100% oxygen without anesthesia, finding that oxygen alone had no impact on either circulating β-HB or glucose (*Figure 1—figure supplement 3*).

## β-HB loss contributes to hypoglycemia in neonates exposed to 1.5% isoflurane

Glucose and β-HB are two key circulating metabolic substrates in neonates; a reduced availability of one of these factors might lead to an increased demand for the other. Given that progressive hypoglycemia follows the rapid loss of β-HB, we hypothesized that this reduction in β-HB may contribute to a subsequent acceleration of glucose depletion. To explore this possibility, we tested whether β-HB supplementation could attenuate the loss of circulating glucose or whether exogenous glucose could prevent β-HB loss. Intraperitoneal (IP) injection with 2 g/kg of glucose at the start of anesthetic exposure substantially raised blood glucose levels at each measured timepoint, as expected (*Figure 1K,L*), but did not attenuate β-HB loss (*Figure 1M,N*). Conversely, an IP bolus of β-HB (20 μmol/g) at the start of isoflurane exposure, which significantly increased blood β-HB levels compared to controls (*Figure 1O,P*), led to a partial but significant attenuation of isoflurane-induced hypoglycemia (*Figure 1Q–R*). At 1 hr, β-HB-treated mice have blood glucose levels not significantly different from baseline (*Figure 1Q*), and at 2 hr, glucose levels in β-HB-treated mice are midway between those of isoflurane-exposed and control-exposed animals (*Figure 1Q*), indicating that β-HB loss contributes to anesthesia-induced hypoglycemia in neonates.

Notably, exogenous β-HB was not sufficient to fully reverse the hypoglycemia in animals exposed to 1.5% isoflurane for 120 min. Interestingly, β-HB is depleted without hypoglycemia in animals exposed to 1% isoflurane for 120 min (*Figure 1*, *Figure 1—figure supplement 1*). Given that lactate is significantly increased by 1.5%, but not 1%, isoflurane exposure, as described above, these data indicate that VA-induced hypoglycemia is predominantly driven by a shift toward anaerobic glucose metabolism, which is significantly less efficient in ATP yield, while maintaining circulating β-HB can attenuate the effects by providing an alternate fuel source.

## Effects of VAs on dietary-induced ketosis

We next considered the possibility that VAs may affect dietary ketosis or fasting-induced ketogenesis. During short-term fasting (see *Figure 1C*), neonates show a significant an increase in blood β-HB by 2 hr. Extending the exposure length, we found there is a similar absolute value increase in β-HB between 60 and 240 min whether neonates are control or 1.5% isoflurane exposed (the isoflurane group at a much lower point at 60 min) (*Figure 2A*). Consistent with this finding, adult animals, which start with low β-HB relative to neonates, show a statistically significant increase by 180 min of either control conditions or exposure to 1.5% isoflurane, with no difference between the two groups (*Figure 2B*). Together, these data show that β-HB induction by fasting is insensitive to VAs, suggesting that the mechanisms underlying the acute impact of VAs on neonatal β-HB may not involve those pathways involved in fasting-induced ketogenesis.

To determine whether isoflurane also effects ketone levels in the setting of dietary ketosis in older mice, we next anesthetized adolescent animals raised on a ketogenic diet to with 1.5% isoflurane (see Materials and methods). These animals have high baseline β-HB, as expected, but their β-HB levels were completely insensitive to 1.5% isoflurane (*Figure 2C,D*). This particular metabolic effect of isoflurane (loss of circulating β-HB) is specific to the neonatal setting.

## β-HB loss in response to isoflurane exposure is specific to neonates

Given the surprising dichotomy of effects on β-HB in neonatal mice compared to β-HB in the setting of adolescent animals in dietary ketosis, we next defined the period of this metabolic sensitivity to isoflurane. Baseline β-HB and glucose concentrations in mice range from age P7 to P30 (*Figure 2E, F*). Baseline β-HB shows a distinct demarcation as a function of age: high levels up until P17, followed by markedly lower levels starting at P19 (*Figure 2F*). In contrast, steady-state glucose levels gradually increase as a function of age from P7 to P30 (*Figure 2F–G*), with no clear shift at P17/P19.

Next, we exposed animals to control conditions or 1.5% isoflurane anesthesia for 1 hr and assessed blood β-HB (*Figure 2I*). Isoflurane exposure resulted in a dramatic depletion of circulating ketones by 1 hr of exposure throughout the neonatal period of P7–P17, with ß-HB reaching a plateau of ~0.5–1 mM in each case. Mice P19 or older show low blood β-HB; isoflurane did not significantly alter levels. Median values by treatment group and age indicate that while isoflurane significantly reduces β-HB in mice up to P17, there is no significant overall effect in animals P19 or older (*Figure 2J*).

As discussed, glucose in P7 neonates is only modestly reduced by 1 hr of 1.5% isoflurane exposure, but markedly low by 2 hr, whereas P30 animals maintain their blood glucose. To define the period of neonatal glucose sensitivity, we exposed mice of various ages to 2 hr of 1.5% isoflurane or control conditions. Isoflurane exposure led to a depletion of glucose during the prenatal period up to post-natal day 13 (*Figure 2K,L*).

## β-HB depletion is uncoupled from sedation and common among volatile anesthetic agents

We previously observed that blood β-HB is depleted in P7 animals exposed to 1% isoflurane for over 2 hr (*Johnson et al., 2019b*). To determine whether β-HB is acutely depleted by non-anesthetizing concentrations of isoflurane, we exposed P7 neonatal mice to 1% isoflurane and assessed circulating metabolites. We found the loss in circulating β-HB is similar in both rate (i.e. half-life of effect) and final effect size in P7 neonates exposed to either 1% or 1.5% isoflurane (*Figure 3A,B*). We further found, remarkably, that the impact of 0.2% isoflurane, the lowest setting on many standard clinical isoflurane vaporizers and well below the EC50, on circulating β-HB was as or more potent than 1.5%, demonstrating a robust uncoupling of the anesthetic effects of isoflurane from its

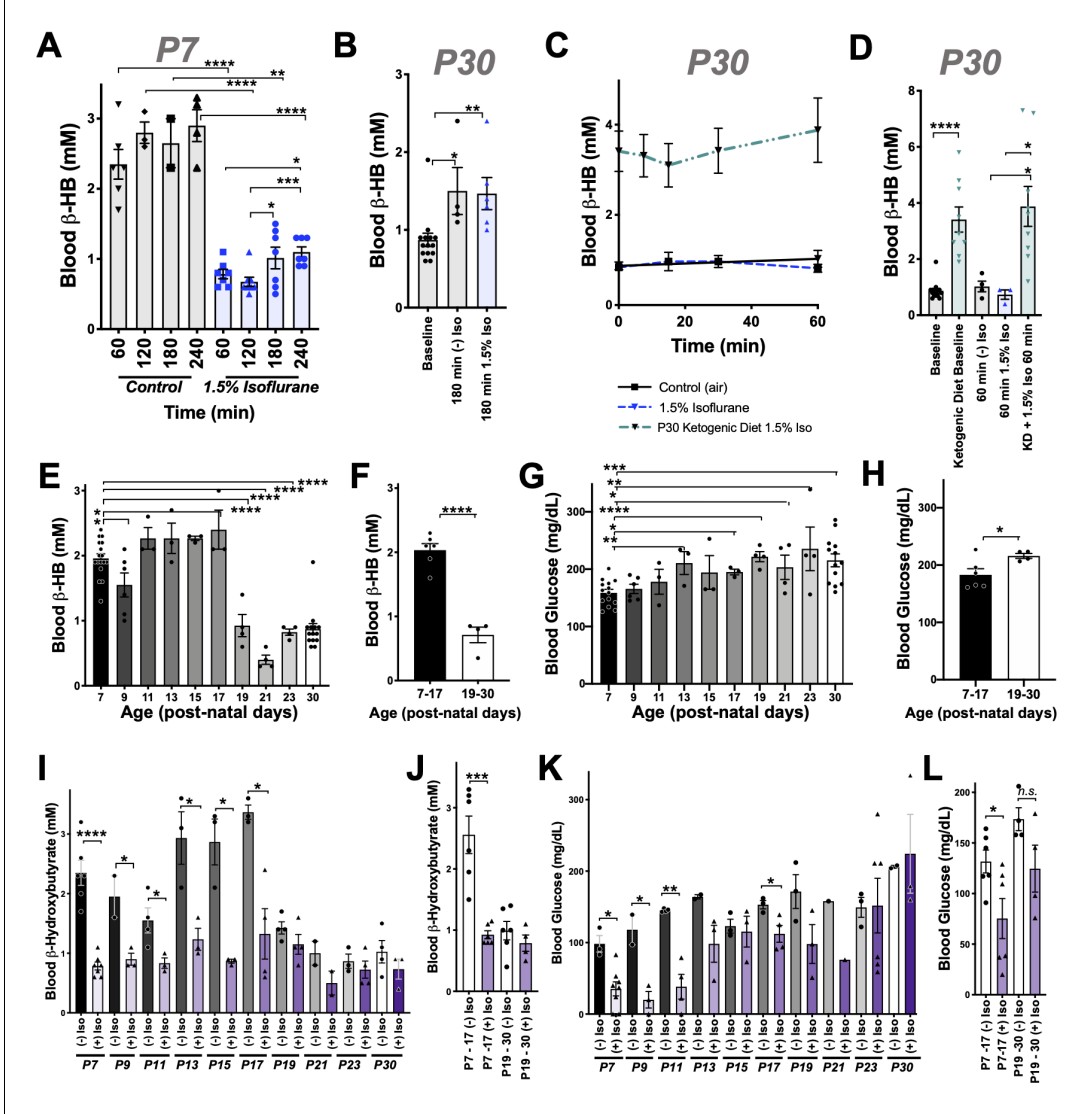

**Figure 2.** Anesthesia sensitivity of ketosis is unique to neonatal ketogenesis. (A) Blood β-HB concentration in P7 mice exposed to 1.5% isoflurane anesthesia for 1–4 hr. One-way ANOVA *p<0.0001; *p<0.05, **p<0.005, ***p<0.005, ****p<0.0001 by pairwise t-test. (B) Blood β-HB concentration in P30 adolescent mice at baseline compared to 180 min of exposure to 1.5% isoflurane or control conditions. One-way ANOVA **p<0.005; *p<0.05, **p<0.005 by pairwise t-test. (C, D) Blood β-HB in P30 mice raised on a ketogenic diet and exposed to 1.5% isoflurane. (C) No significant differences observed at any exposure time compared to baseline within the same group. (D) Bar graphs with individual datapoints from (C), pairwise comparisons to control fed mice and control fed mice exposed to either 1.5% isoflurane or control conditions. One-way ANOVA ****p<0.0001. ****p<0.0001, *p<0.05 by pairwise t-test. Control data also appear in *Figure 1E,F*. (E) Baseline blood β-HB levels in mice as a function of age. One-way ANOVA ****p<0.0001. Comparisons to P7: *p<0.05, ****p<0.0001 by pairwise t-test. (F) Median blood β-HB values by age compared between post-natal periods P7–P17 and P17–P30 ages. ****p<0.0001 by pairwise t-test. (G) Baseline blood glucose levels in mice as a function of age. One-way ANOVA **p<0.005. Comparisons to P7: *p<0.05, **p<0.005, ***p<0.0005, ****p<0.0001 by pairwise t-test. (H) Median blood glucose values by age compared between post-natal periods P7–P17 and P17–30 ages. *p<0.05 by pairwise t-test. (I) Pairwise comparisons of blood β-HB in mice exposed to 1 hr of 1.5% isoflurane or control conditions at various post-natal ages. One-way ANOVA ****p<0.0001. Pairwise comparisons by treatment at each age: *p<0.05, ****p<0.0001 by pairwise t-test. Treatments non-significantly different where p-values not indicated. (J) Median β-HB values in each age and treatment group from (I) grouped by post-natal periods P7–P17 and P17–P30. One-way ANOVA ****p<0.00001, ***p<0.0005 by pairwise t-test. (K) Pairwise comparisons of blood glucose in mice exposed to 2 hr of 1.5% isoflurane or control conditions at various post-natal ages. One-way ANOVA ****p<0.0001. Pairwise comparisons by treatment at each age: *p<0.05, **p<0.005 by pairwise t-test. Treatments non-significantly different where p-values not indicated. (L) Median β-HB values in each age and treatment group in (K) grouped by post-natal periods P7–P17 and P17–P30. One-way ANOVA **p<0.005. Pairwise comparison **p<0.005 by pairwise t-test. (A–L) In all graphs, each datapoint represent values derived from an individual animal, with the exception of F, H, J, and L panels, where individual datapoints represent the mean values at different ages, as indicated. In the P30 datasets, multiple timepoints were collected per animal using the tail-prick method, whereas all neonate datapoints represent one animal with no repeat measurements (see Materials and methods).

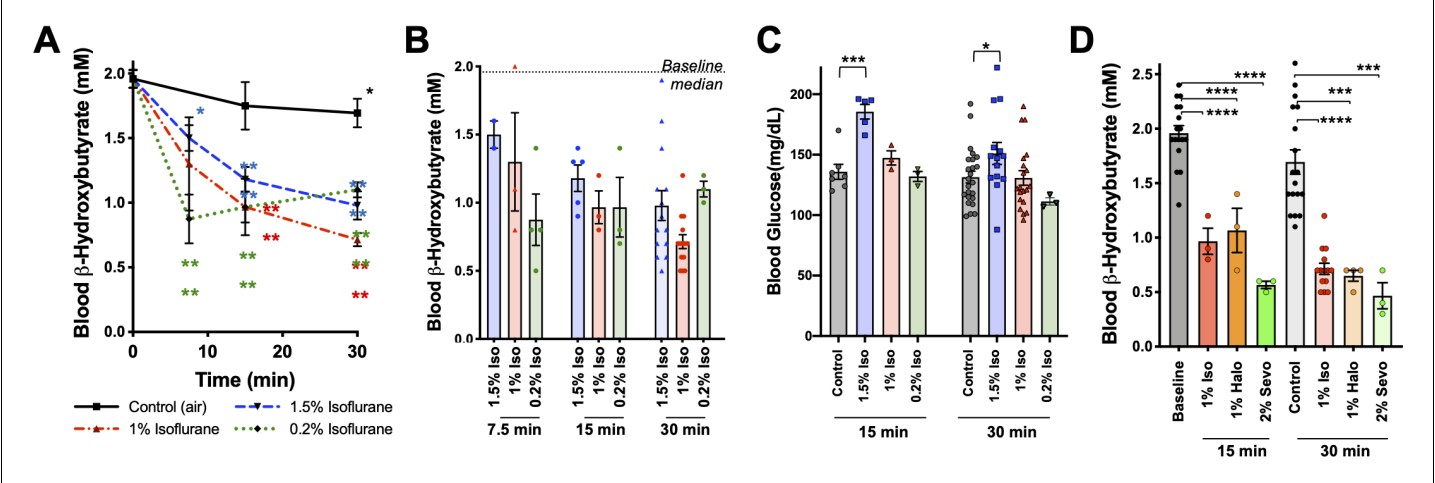

**Figure 3.** Metabolic effects of volatile anesthetics are uncoupled from sedation and properties common to multiple VA compounds. (**A**) Blood β-HB levels in P7 mice exposed to varying levels of isoflurane as a function of time exposed. 1.5% and baseline as in *Figure 1C* shown for comparison. Pairwise comparisons versus baseline *p<0.05, ****p<0.0001 by pairwise t-test. p-values color coded to indicate exposure condition. (**B**) Blood β-HB levels in P7 mice exposed to varying concentrations of isoflurane for 7.5, 15, 30, or 60 min. Median values in baseline and 60 min control treated groups indicated by horizontal lines. (**C**) Blood glucose levels in P7 mice exposed to varying concentrations of isoflurane for 15 or 30 min. ***p<0.0005, *p<0.05 by pairwise t-test. (**D**) Blood β-HB levels in P7 mice exposed to sub-anesthetic concentrations of isoflurane, halothane, or sevoflurane for 15 or 30 min. ****p≤0.0001, ***p=0.0003 by pairwise t-test.

impact on neonatal circulating β-HB. Critically, the lower concentrations of isoflurane (1% and 0.2%) did not cause the transient increase in blood glucose seen in the 1.5% isoflurane-exposed animals, also uncoupling this transient glycemic event from the loss of circulating β-HB (*Figure 3C*).

To determine whether the metabolic effects we observed are common among chemically and structurally distinct VAs, or specific to isoflurane, we tested the impact of halothane and sevoflurane. As with sub-anesthetic concentrations of isoflurane, sub-anesthetic doses of both halothane (1%) and sevoflurane (2%) rapidly reduced β-HB in P7 mice (*Figure 3D*) (see *Johnson et al., 2019b* for sevoflurane EC50).

## Brief exposure to isoflurane impairs fat metabolism

Mammalian milk is high in fatty acids and low in carbohydrates and, accordingly, neonatal animals rely heavily on fat metabolism (*Mitina et al., 2020*). Fatty acid oxidation (FAO) provides acetyl-CoA both to supply the tricarboxylic acid (TCA) cycle and to drive hepatic ketogenesis. Ketogenesis occurs in the liver and provides circulating ketone bodies, including β-HB, for utilization by peripheral tissues. Targeted metabolomic analysis of liver tissue demonstrated that β-HB is depleted in liver by 30 min of exposure to low doses (1% and 0.2%) isoflurane, consistent with a hepatogenic origin of the β-HB-depleting effects of VAs (*Figure 4A*).

Given the importance of FAO in driving neonatal metabolism, including ketone synthesis, we next assessed whether VAs have an impact on overall free fatty acid (FFA) levels in plasma and livers of P7 neonates exposed to 1% isoflurane or control conditions for 30 min. We focused on low-concentration (here using 1%) isoflurane, rather than 1.5%, in this assay, and much of the remainder of this work is to avoid both the glycemic impact and deep sedation of 1.5% isoflurane (see *Figure 1*, *Johnson et al., 2019b*). This allows for isolation of the β-HB phenomena, avoiding these other effects of VAs. One percent isoflurane had no impact on plasma FFAs (*Figure 4B*), and liver FFAs trend upward (*Figure 4C*), indicating that a short exposure to VAs does not impair uptake or distribution of plasma and liver FFAs.

Next, we performed acyl-carnitine profiling to determine whether VAs impact FAO. FFAs are activated by covalent linkage with coenzyme A to form acyl-CoAs (*Grevengoed et al., 2014*). Acyl-CoAs are then conjugated to carnitine via carnitine palmitoyltransferase-1 (CPT1), allowing FAs to enter the mitochondria through the carnitine shuttle (*Longo et al., 2006*). Carnitine shuttling is necessary for long-chain FA transport across the membrane, while short chains can enter by diffusion.

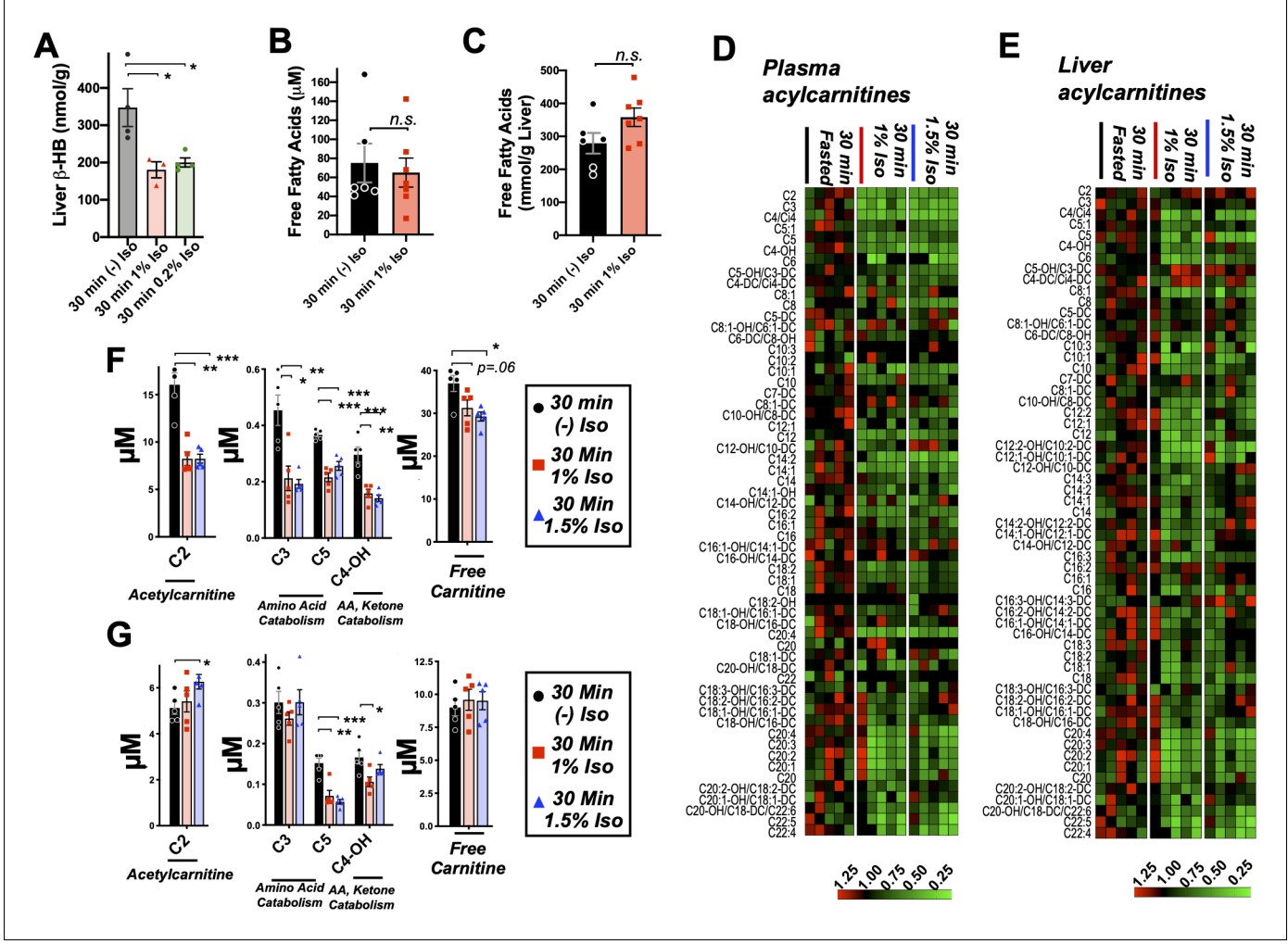

**Figure 4.** Brief exposure to isoflurane impairs fatty acid metabolism. (A) β-HB concentration in whole liver of P7 mice exposed to 30 min fasting (n = 4), 1% isoflurane (n = 3), or 0.2% isoflurane (n = 4). ANOVA p<0.05, *p<0.05 by pairwise t-test. (B) Free fatty acid concentrations in blood of P7 neonatal mice exposed to 30 min of fasting (n = 6) or 1% isoflurane (n = 7). Not significantly different by pairwise t-test. (C) Free fatty acid concentrations in liver of P7 neonatal mice exposed to 30 min of fasting (n = 6) or 1% isoflurane (n = 7). Not significantly different by pairwise t-test. (D) Profiling of plasma acyl-carnitines in P7 neonatal mice exposed to 30 min of fasting, 1% isoflurane, or 1.5% isoflurane, n = 5 per treatment group (each column is one animal). Heat map rows (individual acyl-carnitine species) normalized to 30 min fasted control group median values, with relative levels indicated by color map, below. (E) Profiling of liver acyl-carnitines in P7 neonatal mice exposed to 30 min of fasting, 1% isoflurane, or 1.5% isoflurane, n = 5 per treatment group (each column is one animal). Heat map rows (individual acyl-carnitine species) normalized to 30 min fasted control group median values. Relative levels indicated by color map, below. (F) Plasma concentrations of major acyl-carnitine species C2, C3, C5, and C4-OH, from (D), and free carnitine. *p<0.05, **p<0.005, and ***p<0.0005 by pairwise t-test. Treatment group as indicated by color and datapoint shape indicated in legend. (G) Liver concentrations of major acyl-carnitine species C2, C3, C5, and C4-OH, from (E), and free carnitine. (A–G) *p<0.05, **p<0.005, and ***p<0.0005 by pairwise t-test. Treatment group as indicated by color and datapoint shape indicated in legend.

We found that plasma acyl-carnitines are broadly reduced by both 1.5% and 1% isoflurane (*Figure 4D*), with reductions observed in the majority of species detected. Liver acyl-carnitines are also broadly depleted by both concentrations of isoflurane (*Figure 4E*).

Larger acyl-carnitine species arise exclusively from FAO, while lower molecular weight acyl-carnitines can result from FAO, amino acid catabolism (C3, C5, and C4-OH), or ketone catabolism (C4-OH). C3, C5, and C4-OH were all significantly reduced in plasma, with no difference between 1% and 1.5% isoflurane (*Figure 4F*). These data are consistent with isoflurane impacting FAO broadly, such as at the CPT1 reaction, rather than through impairing a specific FFA generating pathway or alternate acyl-carnitine precursor pathway, such as amino acid catabolism. The lack of a dose dependency further supported our use of 1% isoflurane in subsequent metabolic studies.

C2, acetylcarnitine, is the product of the conjugation of free carnitine with acetyl-CoA and reflects overall acetyl-CoA pools. High mitochondrial acetyl-CoA is reflected by increased C2 and plays a role in inhibition of FAO by driving malonyl-CoA generation by acetyl-CoA carboxylase (ACC)

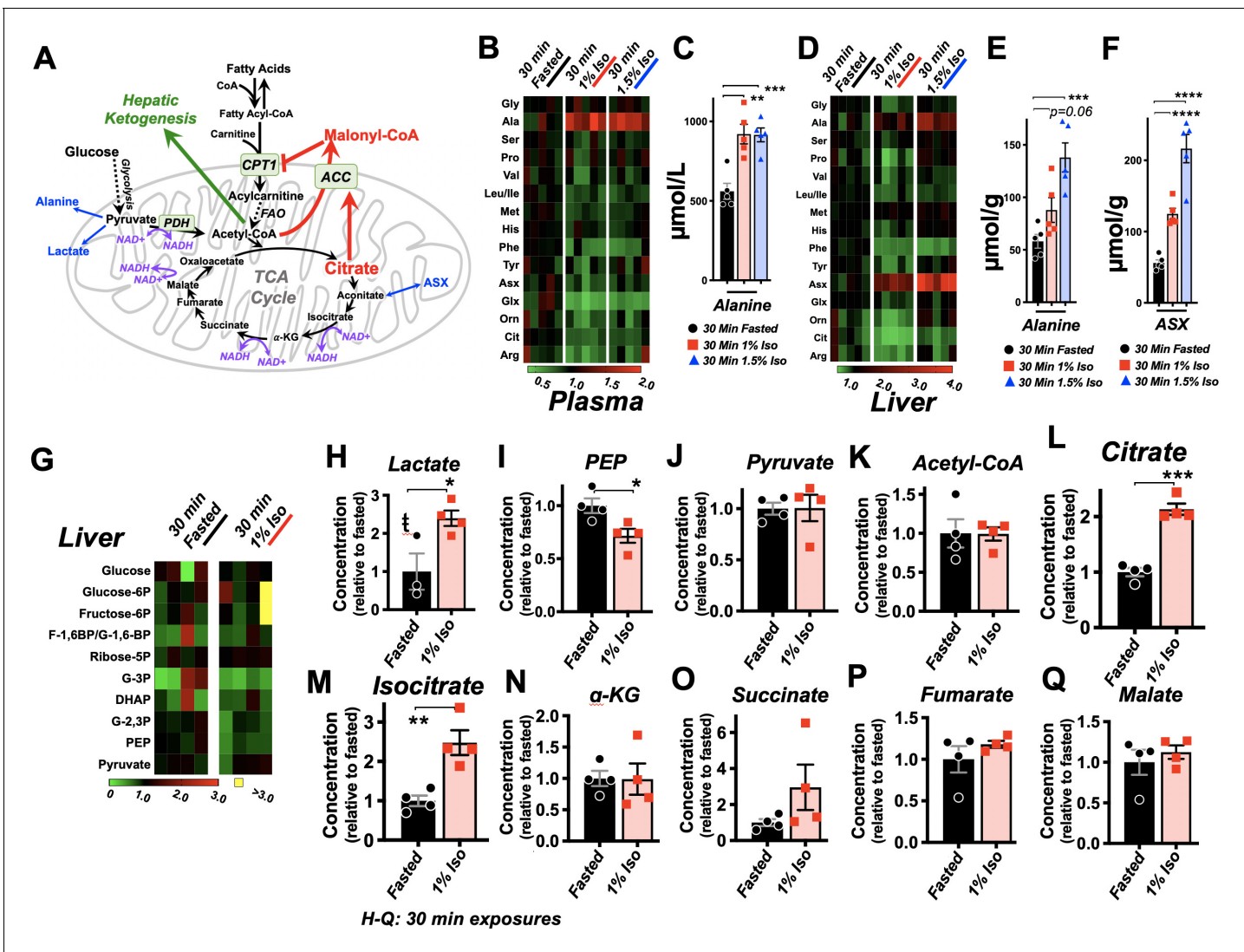

**Figure 5.** Isoflurane exposure leads to cataplerosis and citrate accumulation. (A) Schematic of mitochondrial metabolism of glucose and fatty acids. The cataplerotic amino acids alanine and aspartate/asparagine (ASX) are generated when TCA cycle flux is impaired, and lactate is generated under conditions where glucose entry into the TCA cycle is disrupted (blue text). Citrate plays a key role in mediating feedback inhibition by activating acetyl-CoA carboxylase (ACC) to generate malonyl-CoA, which is an inhibitor of CPT1. CPT1 activity is necessary in order to enable entry of fatty acids into the mitochondria for fatty acid oxidation (FAO). Multiple steps of the TCA cycle consume NADH and are inhibited by NAD+ (purple). (B) Profiling of plasma amino acids in P7 neonatal mice exposed to 30 min of 1% isoflurane, 1.5% isoflurane, or control conditions. Columns represent individual animals, with each metabolite normalized to the 30 min fasted group. (C) Bar graphs of alanine data from (B). ANOVA ***p<0.0005. **p<0.005, ***p<0.0005 by pairwise t-test. (D) Profiling of liver amino acids in P7 neonatal mice exposed to 30 min of 1% isoflurane, 1.5% isoflurane, or control conditions. Columns within the heat map represent individual animals, with each metabolite normalized to the 30 min fasted group. (E) Bar graphs of alanine data from (D). ANOVA ***p-value<0.0005. ***p<0.0005 by pairwise t-test. (F) Bar graphs of aspartate/asparagine data from (D). ANOVA ****p-value<0.0001. ****p<0.0001 by pairwise t-test. (G) Profiling of glycolysis intermediates in P7 neonatal mice treated exposed to 30 min 1% isoflurane or control conditions. Columns within the heat map represent individual animals; each row (metabolite) is normalized to the 30 min fasted group. (H–Q) Lactate (H), PEP (I), pyruvate (J), and TCA cycle intermediates in liver of P7 neonatal mice treated exposed to 30 min 1% isoflurane or control conditions (K–Q). *p<0.05, **p<0.005, ***p<0.0005 by pairwise t-test. ‡ - one outlier in the control group (value 6.61) detected by Grubbs test and removed (a = 0.1) (see *Figure 5—figure supplement 1*).

The online version of this article includes the following figure supplement(s) for figure 5:

**Figure supplement 1.** Additional glycolysis intermediates in 1% isoflurane-exposed neonates.

(*Grevengoed et al., 2014*; *Kiens, 2006*; *Lopaschuk and Gamble, 1994*) (see diagram in *Figure 5A*), which exists in both cytoplasmic (ACC1) and mitochondrial (ACC2) isoforms (*Abu-Elheiga et al., 2000*). While plasma C2 was reduced by roughly 50% by 30 min of exposure to iso-flurane in a dose-independent manner (*Figure 4F*), hepatic C2 was significantly increased by 1.5% isoflurane (*Figure 4G*), indicating that the mitochondrial acetyl-CoA pool in isoflurane-exposed liver is increased over controls. Free carnitine was also significantly reduced in plasma, whereas hepatic-free carnitine was unchanged (*Figure 4F,G*).

## Volatile anesthetics acutely impair the TCA cycle

VAs have been shown to directly inhibit the activity mitochondrial electron transport chain complex I (NADH ubiquinone oxidoreductase). Since this enzymatic complex consumes NADH, inhibition can increase the ratio of reduced nicotinamide adenine dinucleotide, NADH, versus the oxidized form, NAD+ (*Brunner et al., 1975*; *Gellerich et al., 1999*). Three enzymatic reactions in the TCA cycle are directly regulated by this redox pair, and increased NADH/NAD+ inhibits TCA cycle flux (*Tretter and Adam-Vizi, 2005*; *Liu et al., 2018*; *Martínez-Reyes and Chandel, 2020*). TCA cycle impairment can lead to cataplerosis, the removal of TCA cycle intermediates via production of amino acids to prevent mitochondrial matrix accumulation of TCA cycle intermediates. It can also impair pyruvate entry into the TCA cycle, with a concomitant increase in lactate production. Consistent with these data, plasma and liver amino acid profiling, which was obtained with the acyl-carnitine data, revealed a specific increase in the anaplerotic/cataplerotic amino acids alanine and asparagine/aspartate (indistinguishable by the mass-spectrometry method) in liver and alanine in plasma (*Figure 5B–F*); levels of other amino acids decreased. These data provide strong evidence of cataplerosis in the face of impaired TCA cycle function in isoflurane-exposed animals (*Ratnikov et al., 2015*; *Owen et al., 2002*).

To directly assess whether isoflurane exposure leads to TCA cycle perturbations, we next performed targeted metabolomics of TCA cycle and glycolytic metabolites from liver of P7 animals exposed to 30 min of 1% isoflurane or control conditions. While glycolytic intermediates were largely unchanged (*Figure 5G*, *Figure 5—figure supplement 1*), lactate was significantly increased, additional evidence of a TCA cycle backup (*Figure 5*). In glycolysis, only phosphoenolpyruvate was significantly changed (reduced), though the implications of this finding are unclear (*Figure 5I*). Pyruvate was unchanged, and the majority of TCA cycle intermediates show only non-significant trends upward. However, critically, citrate and isocitrate showed a striking and significant elevation in the 1% isoflurane-exposed group, with citrate increased 100% by isoflurane exposure (*Figure 5J–Q*). These data demonstrate that even brief exposure to the 1% isoflurane results in marked, yet specific, changes to hepatic TCA intermediates in P7 neonates.

## Mechanism of VA-induced β-HB depletion in neonates

In addition to driving lactate increases and cataplerosis, accumulation of the TCA cycle intermediate citrate has been shown to regulate FAO through citrate-mediated activation of ACC. Acetyl-CoA availability and high citrate drive ACC activity; ACC produces the potent CPT1 inhibitor and fatty acid synthesis precursor malonyl-CoA, providing a FAO rheostat linked to citrate and acetyl-CoA levels (*Muoio, 2014*) (see *Figure 5A*). In normal conditions, this rheostat provides a switch between fat metabolism and synthesis linked to energetic status of the TCA cycle and acetyl-CoA. Considering together the impact of isoflurane on acyl-carnitines and citrate, we next considered the possibility that TCA cycle inhibition, accumulation of citrate, production of malonyl-CoA, and subsequent inhibition of FAO at CPT1 may be driving the VA-induced depletion of β-HB in P7 neonates. Consistent with this model, targeted metabolomic analysis confirmed that very low-dose 0.2% isoflurane for 30 min, which depletes β-HB (see *Figure 3*), results in a significant increase in hepatic malonyl-CoA in P7 neonates (*Figure 6A*).

Next, we assessed whether blocking FAO through inhibition of CPT1 could lead to a depletion of blood β-HB in neonates. We administered 5 mg/kg etomoxir, a potent and irreversible pharmacologic inhibitor of CPT1, or 100 μmol/kg malonyl-CoA, the endogenous inhibitor generated by ACC, to P7 neonatal pups by IP injection and assessed circulating β-HB following treatment. Strikingly, both etomoxir (*Figure 6B*) and malonyl-CoA (*Figure 6C*) led to a rapid and robust depletion of blood β-HB levels, similar to VA exposure. These data clearly demonstrate that acute FAO inhibition

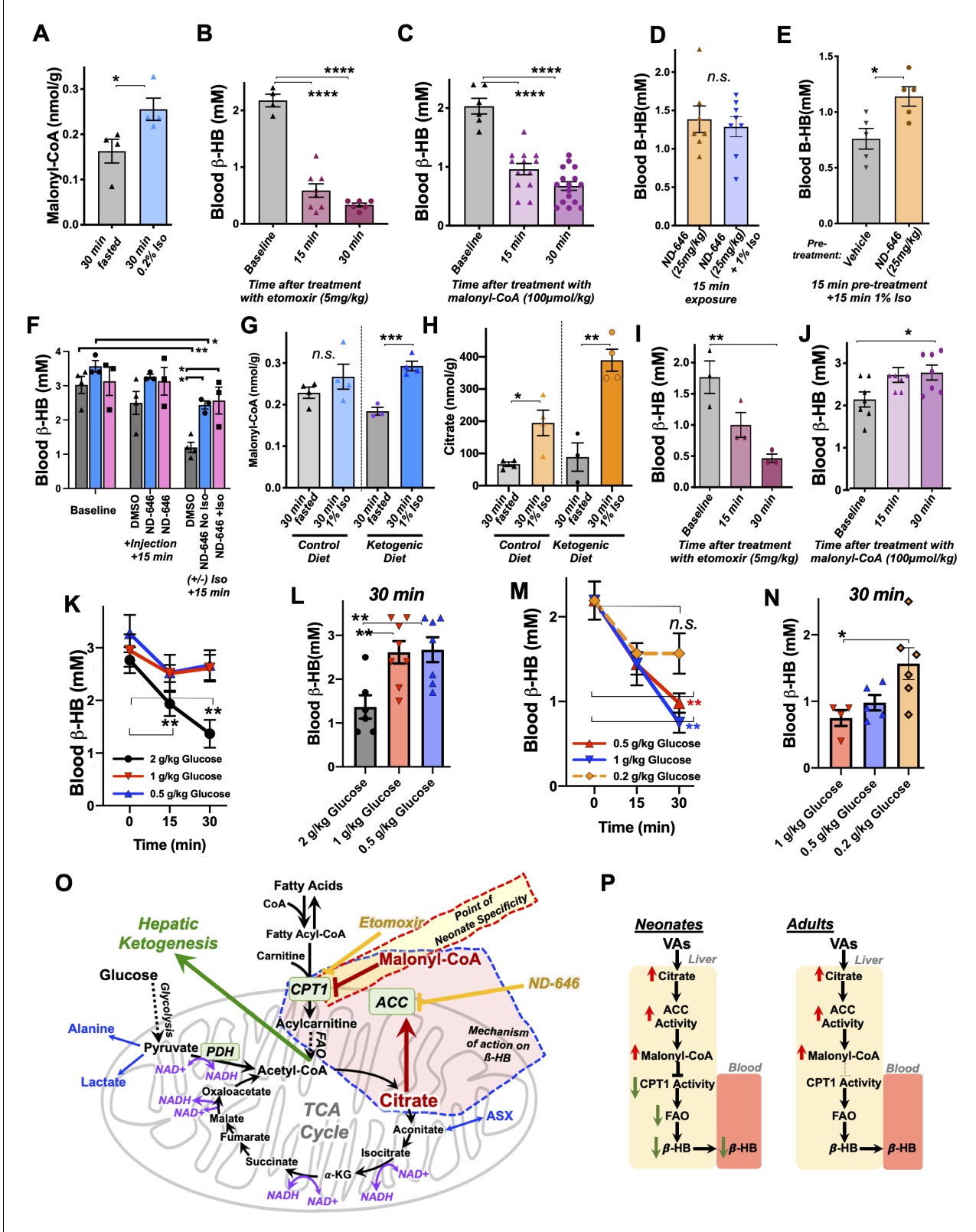

**Figure 6.** Mechanism of β-HB depletion in neonates. (**A**) Hepatic malonyl-CoA concentrations in P7 neonates exposed to 30 min of 0.2% isoflurane or control conditions. *p<0.05 by pairwise t-test. (**B**) Blood β-HB in P7 neonatal mice treated with 5 mg/kg etomoxir by IP injection. ****One-way ANOVA p<0.0001, ****p<0.0001 by pairwise t-test. (**C**) Blood β-HB in P7 neonatal mice treated with 100 µmol/kg malonyl-CoA by IP injection. ****One-way ANOVA p<0.0001, ****p<0.0001 by pairwise t-test. (**D**) Blood β-HB in P7 neonatal mice treated with 20 mg/kg ND-646 followed by 15 min exposure to

*Figure 6 continued on next page*

*Figure 6 continued*

1% isoflurane or control conditions. n.s. – not significant. (**E**) Blood β-HB in P7 neonatal mice treated with ND-646 or vehicle solution 15 min prior to a 15 min exposure to 1% isoflurane. *p<0.05 by pairwise t-test. (**F**) Blood β-HB in P16 neonatal mice at baseline, at 15 min following treatment with 10% DMSO in PBS or 2.5 mg/kg ND-646 in 10% DMSO/PBS, and then after a subsequent 15 min exposure to control conditions or 1% isoflurane. Blood β-HB was reduced by injection with ND-646 over this time period, but ND-646 significantly attenuated the impact of isoflurane. *p<0.05 by pairwise t-test. (**G**) Hepatic malonyl-CoA in P30 animals raised on control or ketogenic diet and exposed to 30 min of 1% isoflurane or control conditions. ***p<0.0005 by pairwise t-test. (**H**) Hepatic citrate in P30 animals raised on control or ketogenic diet and exposed to 30 min of 1% isoflurane or control conditions. *p<0.05, **p<0.005 by pairwise t-test. (**I**) Blood β-HB in P30 mice raised on a ketogenic diet treated with 5 mg/kg etomoxir by IP injection. *One-way ANOVA p<0.05; **p<0.01 by pairwise t-test. (**J**) Blood β-HB in P30 mice raised on a ketogenic diet treated with 100 μmol/kg malonyl-CoA by IP injection. *One-way ANOVA p<0.05; *p<0.05 by pairwise t-test. (**K**) Blood β-HB in P30 mice raised on a ketogenic diet and treated with 2, 1, or 0.5 g/kg glucose by IP injection. **p<0.05 by pairwise t-test, comparison to baseline (t = 0). (**L**) Bar graph of 30 min data in (**J**) to show individual datapoints. O*one-way ANOVA p-value<0.005; **p<0.005 by pairwise t-test. (**M**) Blood β-HB in P7 mice treated with 1, 0.5, or 0.2 g/kg glucose by IP injection. **p<0.05 by pairwise t-test, comparison to baseline (t = 0). (**N**) Bar graph of 30 min data in (**L**) to show individual datapoints. *One-way ANOVA p-value<0.05; *p<0.05 by pairwise t-test. (**O**) Schematic of the metabolic processes underlying the effects of VAs with targets of relevant pharmacologic agents indicated. (**P**) Model of VA action on circulating β-HB in neonates and the differential effects in P30 mice.

The online version of this article includes the following figure supplement(s) for figure 6:

**Figure supplement 1.** ETC CI and the metabolic response to VAs.
**Figure supplement 2.** Acyl-CoA versus acyl-carnitine changes.
**Figure supplement 3.** Impact of exogenous beta-hydroxybutyrate on anesthesia-induced neurotoxicity.
**Figure supplement 4.** Point-of-care meter performance.

leads to depletion of blood ß-HB in neonates, consistent with VAs impacting circulating β-HB through the citrate/ACC/malonyl-CoA/CPT1 pathway.

Finally, to directly assess causality of this pathway in this metabolic effect of VAs, we treated P7 neonatal animals with the ACC inhibitor ND-646 prior to 1% isoflurane exposure to determine whether inhibition of the enzyme responsible for malonyl-CoA production could attenuate the depletion of β-HB induced by VA exposure. ND-646 acts by binding to ACC and mimicking an inhibitory phosphorylation site, which prevents enzyme dimerization; however, already dimerized ACC is resistant to inhibition, and citrate activates ACC through other mechanisms, so a full rescue of β-HB levels in this instance was not anticipated (*Park et al., 2002*; *Lee et al., 2018*; *Svensson et al., 2016*; *Li et al., 2019*). We performed this experiment using two slightly different paradigms: (1) comparison of animals treated for 15 min with ND-646 with or without simultaneous exposure to 1% isoflurane and (2) pre-treatment of animals for 15 min with either ND-646 or vehicle solution followed by exposure to 1% isoflurane (i.e. a pre-treatment with ND-646). In each case, mice from individual litters were equally distributed between treatments. In paradigm 1, ß-HB levels were not lower in mice treated with ND-646 then exposed to isoflurane when compared to animals treated with ND-646 but not exposed to isoflurane (*Figure 6D*). In paradigm 2, β-HB was significantly higher in animals pre-treated with ND-646 compared to those pre-treated with vehicle solution (*Figure 6E*).

We observed that the recommended buffer used to deliver ND-646 (10% DMSO/40% PEG400/5% Tween-80/1X PBS) lowered baseline β-HB in these experiments, so performed one final treatment by diluting ND-646 in 100% DMSO directly into 1× PBS at the moment before injection (see Materials and methods). For this experiment, we treated P16 animals, the age where β-HB is highest (see *Figure 2*). We used three litters, evenly distributing animals between treatments. In this experiment, we found that β-HB levels dropped slightly over time in the ND-646-treated mice, but that treatment again significantly attenuated the drop caused by exposure to isoflurane (*Figure 6F*). Taken together, these data provide causal evidence that ACC driven generation of malonyl-CoA and a resulting inhibition of FAO drives the loss of blood in VA-exposed neonatal mice (for reference, see model in *Figure 6O*).

## Causes of neonatal specificity

Given these data, we next sought to further define the mechanisms of the neonatal specificity of the β-HB depletion resulting from VA exposure. We first performed targeted metabolomic analysis of malonyl-CoA and TCA cycle intermediates in adult (P30) mice fed a ketogenic diet, which we had found have high β-HB levels insensitive to VAs (see *Figure 1*). We postulated that adult mice may have higher basal levels of malonyl-CoA or citrate relative to neonates, resulting in a relative

insensitivity to changes induced by VAs. Adult mouse malonyl-CoA levels are not significantly different compared to neonates at baseline (*Figure 6G* – see legend), while exposure to 30 min of 1% isoflurane results in a significant increase in malonyl-CoA in livers of adult mice on a ketogenic diet (*Figure 6F*). Furthermore, hepatic citrate concentrations trend lower in adults compared to neonates, with no difference between diet groups, while 30 min of 1% isoflurane significantly raised citrate in both dietary conditions (*Figure 6H*). These data reveal, unexpectedly, that the impact of isoflurane on hepatic citrate and malonyl-CoA is as robust in P30 animals as in neonates. The metabolic impact of VAs is the same at both ages at this point in the citrate/malonyl-CoA/CPT1 pathway.

Next, we treated ketogenic P30 mice with etomoxir and malonyl-CoA at the same doses used in neonates to determine whether β-HB production is less sensitive to regulation through CPT1 at this age. Treatment with the potent and irreversible CPT1 inhibitor etomoxir resulted in a rapid reduction in circulating β-HB, as seen in neonates (*Figure 6I*). In striking contrast, treatment with malonyl-CoA had no impact on β-HB levels in ketogenic adults, with β-HB trending upward to 30 min. Together with the findings that citrate and malonyl-CoA are robustly increased by isoflurane in adults as in neonates (*Figure 6G–H*), these data show that FAO regulation through CPT1 plays the same overall role in β-HB in ketogenic adults as in neonates, but that adults are insensitive to regulation of FAO by the endogenous inhibitor malonyl-CoA, providing a striking contrast to the impact of malonyl-CoA in neonates.

These findings indicate that the age specificity of blood β-HB depletion in response to exposure to VAs is the result of a rapid metabolic flexibility in neonates not present in adult animals. Based on this, and our model that the regulation of β-HB is mediated by increased citrate in the presence of abundant acetyl-CoA, we reasoned that β-HB levels should be acutely impacted by glucose administration, which feeds acetyl-CoA and the TCA cycle. This response would be the same in both neonates and adults, but neonates should be more sensitive to regulation by glucose compared to adults. Consistent with this model, we found that 2 g/kg IP glucose, but not 1 or 0.5 g/kg, results in a rapid depletion of blood β-HB in P30 ketotic mice (*Figure 6K,L*). In neonates, blood β-HB was still depleted at glucose doses of 1 or 0.5 g/kg, while the effect was attenuated by 0.2 mg/kg (*Figure 6M,N*). Neonates are between 4- and 10-fold more sensitive to acute metabolic changes in glucose availability, strongly supporting our models for the actions of VAs and the age specificity of their effects (*Figure 6O*).

## Role of ETC CI

VAs have been shown to impair ETC function and directly inhibit ETC CI (*Kayser et al., 2001*; *Hanley et al., 2002*; *Bains et al., 2006*; *Kayser et al., 2011*; *Olufs et al., 2018*; *Zimin et al., 2018*). ETC CI inhibition leads to an increased NADH/NAD+ ratio, which can impair TCA cycle flux at the reversible steps where NADH is generated and NAD+ is consumed (see *Figure 6N*). Given these data, we performed various experiments aimed defining the precise role ETC CI inhibition and NADH redox shifts which mediate the effects of VAs, as detailed below:

First, we assessed NADH and NAD+ in liver of control and isoflurane-exposed animals (P7 and P30) through targeted metabolomics, finding no differences in NAD+, NADH, or the ratio between the two (*Figure 6—figure supplement 1*).

The NAD+ precursor nicotinamide riboside (NR) has been found to attenuate multiple outcomes arising from ETC CI impairment in vitro and in vivo, through rescue of NAD redox (*Schöndorf et al., 2018*; *Walker and Tian, 2018*; *Goody and Henry, 2018*; *Felici et al., 2015*; *Cantó et al., 2012*; *Lee et al., 2019*). To further test the role NADH/NAD+ in the rapid depletion of β-HB, we injected P7 neonates with saline or 500 mg/kg NR, a dose reported to acutely increase NAD+ (*Hong et al., 2018*), 30 min prior to exposure to 1% isoflurane (*Figure 6—figure supplement 1*). NR failed to attenuate the loss of β-HB, but, rather, seemed to exacerbate the effect.

Given the caveats of measuring redox molecules, we further considered the role of ETC CI using pharmacologic and genetic approaches. Treatment of P7 neonates with 0.5 mg/kg rotenone led to lactate and glucose changes similar to that seen with 1.5% isoflurane, but β-HB was unchanged (*Figure 6—figure supplement 1*). Lowering the dose to 0.1 mg/kg resulted in no overt effects on blood metabolites by 30 min, while increasing to 5 mg/kg led to an increase in all measured metabolites, including β-HB.

Finally, we assessed β-HB levels in P17 neonatal control and *Ndufs4* (KO) mice. Ndufs4 is a structural/assembly component of ETC CI, and mitochondrial CI-driven respiration is reduced in *Ndufs4*

(KO) animals (*Johnson et al., 2020*; *Kayser et al., 2016*; *Johnson et al., 2013*; *Quintana et al., 2012*; *Kruse et al., 2008*). This age was chosen in order to take advantage of the fact that neonatal mice still had high blood β-HB (see also *Figure 2*), while *Ndufs4* (KO) mice can be readily identified by a unique hair-loss phenotype. To our surprise, baseline β-HB levels were significantly higher in *Ndufs4* (KO) neonates than in their control littermates (*Figure 6—figure supplement 1*). Together, these data suggest that ETC CI is not the direct target of VAs mediating the acute β-HB effect in neonate, but may contribute to the increased lactate observed in VA-exposed animals.

## Discussion

In this study, we have identified rapid depletion of circulating β-HB as a previously unreported metabolic consequence of VA exposure, defined the age specificity of this finding, determined it is fully uncoupled from sedation, and elucidated the both the mechanism underlying β-HB depletion and the underpinnings of the neonate specificity. Our data provide important new insights into the impact of VAs in neonates and a particularly sensitive population. Ketone bodies are critical metabolites in neonatal and infant mammals, accounting for as much approximately 25% of basal neonatal energy consumption, while ketone consumption rates in neonate brain are four times, and infants five times, that of adults (*Bougneres et al., 1986*; *Kraus et al., 1974*). In the process, we have also demonstrated that short-term exposure to low-dose VAs results in substantial perturbations to hepatic metabolism, including leading to elevated citrate and malonyl-CoA, even in adult mice and irrespective of diet. These data have shed fresh light on the physiologic effects of VAs, but significant questions remain unanswered, which will require further study.

### ETC CI

Together, these data suggest that ETC CI inhibition may play a role in mediating some metabolic effects of VAs, such as VA-induced lactate production, but the precise role of ETC CI in β-HB regulation remains uncertain. Tissue specificity in drug actions may play a role in the differences between VAs and rotenone, with VAs preferentially impairing function in β-HB producing, vs consuming, tissues, or differences in the pharmacokinetics/dynamics of inhibition. The precise nature of ETC CI inhibition may also be distinct, with differential metabolic effects confounding the impact of rotenone. The *Ndufs4* (KO) data may indicate that chronic reduction in ETC CI function leads to compensatory increases in β-HB output. Each of these questions will need to be addressed in order to fully understand the role of ETC CI in the metabolic effects of VAs.

### Direct target of VAs

ETC CI may play a key role in the accumulation of citrate and subsequent metabolic changes, as discussed. Aconitase and IDH are responsible for the conversion of citrate to isocitrate and isocitrate to alpha-ketoglutarate, respectively. The energetically costly IDH reaction is most sensitive step of the TCA cycle to regulation by NADH redox (*Gabriel et al., 1986*; *Al-Khallaf, 2017*; *Kim et al., 1999*). Accordingly, the TCA cycle block at this NAD+ consuming step is consistent with altered NADH/NAD+ homeostasis within the mitochondria resulting from ETC CI inhibition. However, our data did not support a model whereby citrate is increased as a result of ETC CI inhibition. It is possible that VAs directly impair mitochondrial TCA cycle enzymes – the lack of detectable NADH redox shifts in liver and failure of rotenone or *Ndufs4* deficiency to mimic VAs seem to support this possibility (*Figure 6—figure supplement 1*). Additional studies are needed to resolve the direct target of VAs in this setting.

While our data do not reveal the identity of the direct target of VAs in this paradigm, we successfully uncoupled sedation from the hepatic citrate/malonyl-CoA/FAO/ß-HB pathway. Moreover, since they occur at such low doses, these off-target effects cannot be avoided by simply turning down the dose of anesthetic. Whether any metabolic effects of VAs in neurons are similarly uncoupled from sedation remains to be determined.

The underpinnings of the age-related change in responsiveness to malonyl-CoA will require further study. The most likely culprit is CPT1, which has three isoforms – CPT1a, CPT1b, and CPT1c (*He et al., 2012*). CPT1a is predominant most tissues, including liver, but is absent from muscle and brown adipose tissue, where CPT1b is the main form; CPT1c is expressed in the brain and appears to play a role in feeding behavior (*Lee et al., 2015*; *Brown et al., 1997*; *Yamazaki et al., 1997*;

*Lavrentyev et al., 2004*; *Price et al., 2002*). Any development-related changes in CPT1 expression, isoform preference, or post-translational modifications modulating activity could lead to the altered sensitivity to malonyl-CoA and would provide intriguing insight into the developmental regulation of FAO and ketone production. Similarly, while genetic defects in CPT1 and CPT2 have been shown to underlie pathogenic responses to VAs, including rhabdomyolysis, hyperkalemia, metabolic acidosis, and even acute renal failure and cardiac arrest (*Benca and Hogan, 2009*; *Wieser et al., 2008*; *Cornelio et al., 1980*), the role of CPT1 has not been directly probed in relation to anesthesia in normal patients or in genetic mitochondrial disease. Further study of CPT1 in these settings seems warranted given its importance in the impact of VAs on neonatal metabolism.

We have shown that a 30 min exposure to isoflurane broadly depletes acyl-carnitines in neonatal liver without reducing total free fatty acid levels (*Figure 3*). However, while the impact VAs on acyl-carnitine production is similar between individual FA species, VAs may differentially impact FAs of different lengths at other steps in their metabolism. Consistent with this possibility, a small panel of long-chain FA acyl-CoA and acyl-carnitines shows that the impact of isoflurane on acyl-CoAs is mixed: VA exposure has no impact on C20:4 and C18 acyl-CoA levels, while C14, C16, and C20 acyl-CoAs are reduced to a degree similar to that of their respective acyl-carnitines (see *Figure 6—figure supplement 2*). Accordingly, while we can confidently state that VAs disrupt acyl-carnitine production and interfere with FAO at this step, the precise impacts of VAs on the metabolic fates of individual FA species and FAO intermediates remain undefined.

## Anesthesia-induced neurotoxicity

Our findings raise the important question of whether the depletion of circulating β-HB contributes to the neurotoxic effects of volatile anesthetic exposure in neonatal animals. In a pilot group, we found that administering 2 μmol/g β-HB by IP injection immediately prior to an anesthesia exposure (4 hr exposure with 2 hr recovery, as we have done previously [*Johnson et al., 2019b*]) significantly reduced the number of cleaved caspase-3-positive nuclei in at least one brain region, cortex (*Figure 6—figure supplement 3*). Further studies should aim to characterize the scope of these effects on markers of damage and neurocognitive outcomes and test alternative ketone-raising intervention strategies. Additional studies aimed at probing the relationship between β-HB and anesthesia-related neurotoxic outcomes in both neonatal and non-neonatal models are warranted.

## Beyond mice

Available evidence indicates that our findings will extend to other rodent models, and possibly to higher mammals, including humans. Neonate-specific regulation of fat metabolism and ketogenesis through malonyl-CoA/CPT1 has been reported in rats and rabbits (*Williamson, 1985*; *Pegorier et al., 1992*; *Prip-Buus et al., 1990*), with a very similar age dependency. Furthermore, levels of hepatic and intestinal CPT1 and HMG-CoA synthase, which is involved in ketogenesis, have been shown to be high during the neonatal period in these rats and rapidly decline at weaning (*Asins et al., 1995*; *Serra et al., 1993*), suggesting that the mechanism of action we have uncovered is similar in these species.

Similarly detailed data are unavailable for human neonatal and pediatric populations, but human neonates are known to be hyperketotic compared to adults (*Laffel, 1999*), and neonatal ketone body turnover has been shown to be as high in neonates as in adults who have been fasting for multiple days (*Bougneres et al., 1986*). These findings underlie estimates that ketone provide as much as 25% of circulating energy requirements in neonatal humans. Further study is needed to define the normal concentrations β-HB in healthy human newborns and infants as a function of age and to assess whether exposure to volatile anesthetics similarly depletes β-HB in human neonatal patients.

## Experimentally important non-anesthetic effects of VAs

Our data have major implications to any research utilizing anesthesia prior to assessing metabolic end points. We have clearly demonstrated that exposure to VAs has a rapid and significant impact on many metabolites including β-HB, citrate, malonyl-CoA, and acyl-carnitines. Some of these extend to adult animals and are likely to occur in other vertebrates, including humans, as well. Our data indicate that great caution should be used when considering the use of VAs in experiments

involving metabolic endpoints, as even brief exposure at low dose can have striking metabolic consequences.

## Materials and methods

### Ethics statement and animal use

All experiments were approved by the Animal Care and Use Committee of Seattle Children's Research Institute (Seattle, WA). Experiments utilize the C57Bl/6 strain, originally obtained from Jackson laboratories (Bar Harbor, ME), or the *Ndufs4* (KO) strain, originally obtained from laboratory of Dr. Richard Palmiter at the University of Washington (*Kruse et al., 2008*). All treatment group assignments were randomized. Animal numbers for each dataset are noted in the associated figure legends. *Ndufs4* (KO) mice were bred by heterozygous mating and genotyped using the Jackson laboratory PCR method. Animals used for *Ndufs4* (KO) P17 data were identified by the hair-loss phenotype that occurs in the *Ndufs4* (KO) animals. *Ndufs4* deficiency is a recessive defect, and heterozygosity results in no reported phenotypes, including no detectable defects in ETC CI activity, so controls for this dataset include both heterozygous and wild-type mice.

Cages were checked for weanlings every 1–2 days. Neonatal animal ages are within a 24 hr window of the indicated age – for example, all 'P7' neonates are 7–8 post-natal days old. P30 animals were between 30 and 35 days old, and P60 animals were between P60 and 65 days old. No differences in any metabolic end points are anesthesia sensitivities were identified within these defined age ranges. In pilot studies, we found no differences between male and female animals. All neonatal experiments were performed on an equal (randomized) mixture of male and female pups. All adolescent/adult experiments were performed on male animals.

When possible, blood point-of-care measures were collected from animals, which were used for tissue collection for metabolite studies, maximizing our replicate numbers for point-of-care data.

All experiments contain data from animals from two or more litters to avoid any litter or parenting effects.

All animals were euthanized by decapitation (neonates) or cervical dislocation followed by decapitation (adults) following animal care regulations.

### Anesthetic exposures and control conditions

We chose anesthetic conditions consistent with standard of care in veterinary medicine and published mouse neonatal anesthesia literature (see *Johnson et al., 2019b*). Isoflurane (Patterson Veterinary, 14043070406), halothane (Sigma, B4388), or sevoflurane (Patterson Veterinary, 14043070506) were provided at concentrations indicated using a routinely maintained and tested isoflurane vaporizer (Summit Anesthesia Solutions, various models) at a flow rate of 3–4 liters/min through a humidifier in-line. Vaporizers were routinely calibrated by a commercial service, and performance was monitored using an in-line volatile anesthetic concentration sensor (BC Biomedical AA-8000 analyzer). One hundred percent oxygen was used as the carrier gas, as detailed. The plexiglass exposure chamber and humidifier were pre-warmed to 38°C and maintained at this temperature throughout the exposure using a circulating water heating pad (Adroit Medical, HTP-1500); the temperature of the heating pad was verified using a thermometer. 'No anesthesia' controls were treated identically to isoflurane-exposed animals – this included removal from parents at neonatal ages and fasting, with no access to water, in a normal mouse cage on a 38°C heating pad for the duration of the 'control' exposure for all 'control' treated animals.

### Animal diets

Breeders (parents of experimental neonatal mice) were fed PicoLab Lab Diet 5053; control fed adult mice were fed PicoLab Diet 5058.

To avoid weaning stress and associated weight loss, ketogenic adult mice were gradually acclimated to the ketogenic diet (Envigo, Teklad TD.96335) starting at weaning (P21) using the following protocol: 3 days (starting at weaning) on a 50/50 mix (by weight) of ketogenic diet and ground normal mouse diet (PicoLab Diet 5058), followed by 3 days of 75/25 ketogenic/normal, then 3 days of 85/15. Finally, mice were moved to a 95% (by weight) ketogenic diet. Mice were used for experiments 3–5 days after this final dietary change.

## Point-of-care blood data

Longitudinal collection of blood data is physically impossible in P7 neonatal mice due to their extremely small size. Each blood value measurement therefore represents a single animal euthanized at the timepoint designated. Animals were rapidly euthanized by decapitation, and blood was analyzed immediately. Point-of-care blood data (glucose, β-HB, and lactate) collected from animals aged P17 or older were collected using a minimally invasive tail-prick method, with multiple measures taken from the same animals during time-course data collection.

Except where indicated otherwise, blood glucose was measured using a Prodigy Autocode glucose meter (product #51850–3466188), blood β-HB was assessed using a Precision Xtra XEGW044 meter with β-HB assay strips, and blood lactate was measured using Nova Biomedical assay meter (Product #40828). Each of these meters was assessed for accuracy and precision across relevant blood metabolite concentration ranges, and all were found to be sufficiently reliable for the comparisons in this study (see *Figure 6—figure supplement 4*). All three meters showed good read to read consistency (low SEM between individual reads of the same concentration) and strong linearity of reads throughout the relevant range. The lactate meter became inaccurate at concentrations higher than 5 mM, but given that samples only fell in this range in extreme treatments, and this limitation would only act to amplify the already significant increases in lactate we report and have no impact on the overall outcome, we did not apply any corrections. None of the findings in this manuscript would be impacted by adjusting to compensate for the variances from actual values of any of these point-of-care meters, so no adjustments were applied, and the biological variance in far outweighs the technical variance of these devices.

## Sample collection and storage

All tissues and blood collected for metabolite analysis were flash-frozen in liquid-phase nitrogen and stored at −80°C, or in dry ice (during shipment), until use. Blood used for metabolite analyses was collected using heparinized syringes (Pro-Vent, 4629 P-2) and either frozen whole (whole-blood) or, for plasma, samples were briefly spun in a set-speed table-top centrifuge (Thermo MySpin6 or similar) to pellet blood cells, and plasma was moved to a new tube and then flash-frozen.

## Blood and tissue metabolite analyses

Free fatty acids were quantified using the Abcam Free Fatty Acid Quantification Kit (Abcam, ab65341), following manufacturer's protocol.

Acyl-carnitines were analyzed by the Duke Molecular Physiology Institute Metabolomics Laboratory (Duke University, Durham, NC), as previously described (*Newgard et al., 2009*; *Shah et al., 2012*). Briefly, samples were cryohomogenized under on dry ice using a cryopulverizer (BioSpec) chilled in liquid nitrogen. Frozen powdered tissue was transferred to a tube on dry ice, weighed in a cold analytical scale (Denver Instruments), and homogenized in 50% aqueous acetonitrile containing 0.3% formic acid, to final concentration of 50 mg/mL homogenate. Plasma samples were mixed with 50% aqueous acetonitrile containing 0.3% formic acid to a final concentration of 50 μL plasma/mL total volume. Samples were shipped to Duke, where targeted quantitative tandem flow injection mass spectrometry (MS/MS) was used to detect of 60 metabolites (45 acyl-carnitines and 15 amino acids). For MS/MS analyses, samples were spiked with a cocktail of heavy-isotope internal standards (Cambridge Isotope Laboratories, MA; CDN Isotopes, Canada) and deproteinated with methanol. Supernatant was dried and esterified with hot, acidic methanol (acyl-carnities) or n-butanol (amino acids). Tandem MS/MS using a Waters TQD (Milford, MA) was used to quantitatively assess acyl-carnitine and amino acid ester concentrations. Samples were tested in random order, and samples were blinded to the metabolomic facility.

Samples for TCA cycle, glycolysis, and related analytes, including malonyl-CoA, were flash-frozen in liquid nitrogen and shipped to Creative Proteomics (Shirley, NY) for processing and analysis. All samples were blinded and run in a random order. Briefly, samples were homogenized in mass-spectrometry grade water at 2 μL/mg using an MM 400 mill mixer for three cycles, 1 min each. Methanol was added to 8 μL/mg raw starting material, and homogenization was repeated. Samples were vortexed, sonicated, and centrifuged to clear insoluble material. Clear supernatant was transferred to new tubs for analysis as follows:

Carboxylic acids analysis: Twenty microliters of each standard solution or each clear supernatant was mixed with 20 µL of an internal standard, 20 µL of 200 mM of 3-NPH solution, and 20 µL of 150 mM of EDC solution. This mixture was allowed to react at 30℃ for 30 min. After reaction, 120 µL of water was added to each solution, and 10 µL of the resultant solutions was injected into a C18 UPLC column to quantitate the carboxylic acids by UPLC-MS.

Cofactors analysis: Twenty microliters of the supernatant of each sample was mixed with 180 µL of an internal standard solution containing isotope-labeled AMP, ATP, NAD, and NADH. Ten microliters of each sample solution or each standard solution was injected into a C18 column to quantitate the cofactors by UPLC-MS.

LC parameters: Mobile Phase A: 5 mM tributylamine buffer, mobile Phase B: methanol. The column temperature was held at 50℃. The efficient gradient was from 15% B to 60% B in 20 min, with a flow rate of 0.25 mL/min. Metabolites are quantified using a Thermo Ultimate 3000 UHPLC coupled with an AB Sciex 4000 QTRAP instrument operated in the mode of multiple-reaction monitoring/MS.

## Blood metabolic hormone analysis

Blood was flash-frozen and sent to Eve Biotechnologies for commercial analysis. Analysis by Eve is performed using MilliPore Multiplex antibody-based arrays with robust in-house quality control measures and calibration curves.

## Pharmacologic agent treatment

All agents were administered as intraperitoneal injection at the doses indicated, with working concentrations set so that injection volumes always equaled 100 µL/10 g mouse weight. ND-646 was manufactured by MedChemExpress and purchased through Fisher Scientific (cat. # 501871896). Malonyl-CoA (cat. # M4263), etomoxir (cat. # E1905), rotenone (cat. # R8875), beta-hydroxybutyrate (cat. # 54965), and glucose (cat. # G7021) were obtained from Sigma. Beta-hydroxybutyrate and glucose were dissolved in 1× phosphate buffered saline (PBS) (Corning, 10010023). The remaining agents, other than ND-646, were dissolved in DMSO (Sigma, D8418) or water to 1000X stocks and diluted to 1× in 1× PBS (Corning, 10010023) before injection. Rotenone was prepared immediately before use, as the higher dose rapidly falls out of solution upon dilution to 1×. ND-646 was dissolved to 2.5 mg/mL in 10% DMSO/40% PEG400/5% Tween-80/1X PBS with final pH adjusted to 7.4, as per manufacturer recommendations, and this mixture was used for injection at 100 µL/10 g to achieve a 25 mg/kg dose. Vehicle treated animals for the ND-646 experiments received this vehicle solution with no ND-646 added. Alternatively (see *Figure 6*), 2.5 mg/mL ND-646 in 100% DMSO was dissolved directly into PBS for a final concentration of 0.25 mg/mL immediately before injection.

## Statistical analyses, power calculations, and replicate numbers

All statistical analyses were performed using GraphPad Prism as detailed in figure legends.

All experiments were initially approached using an n of 4 per group for experiments, which did not have to be run in the same batch (such as point-of-care data from neonates) and an n of 5 for experiments where running all samples together was critical to avoid batch effects (such as acyl-carnitine analyses). The rational for this approach was based on the calculation that four replicates per group is needed for an 80% power to detect changes of 30% or greater in groups with a 15% sigma. A fifth replicate was included for experiments with batch concerns, so that if a sample was lost in processing the dataset would still be complete.

For animal point-of-care blood data experiments, the total n was modified as necessary and justified to minimize animal use or resolve strong trends. Specifically, control treatment conditions were pooled where possible (for nearly all anesthesia exposures), increasing the overall power of each comparison to controls; the desired replicate number was reduced to three in instances where changes were already highly significant upon collecting three samples; additional replicates were included when strong trends were apparent after four replicates were collected; as many blood values as possible were collected from mice used for tissue collection/metabolomics datasets to maximize their purpose, leading to increased replicate numbers in additional instances. Pooled control (t = 0) data was not used for treatments in *Figure 6*, which probed the model of anesthetic action (i.e. malonyl-coa administration, etomoxir administration, etc). In these cases, data shown represent

only mice from the same litters split amongst the treatments per time. Including pooled control data would only further enhance the already statistically significant data.

## Cleaved caspase-3 (supplemental) staining and quantification

Mouse brains were rapidly extracted after euthanasia and placed in ice-cold 3.8% paraformaldehyde in 1× PBS for 24 hr. Following fixation, brains were moved to a cryoprotectant solution (30% sucrose, 1% DMSO, 100 µM glycine, 1× PBS, 0.45 µm filtered, pH 7.5) and stored for 48–72 hr. Cryoprotected tissues were frozen in OCT media (Tissue-Tek OCT compound, Sakura 0004348–01) in cryoblock holders on dry ice and stored at −80°C until sectioned.

Cryoblocks were cut at 50 µm thickness using a Leica CM30505 cryostat set at −40°C. Slices were moved to 1× PBS and stored at 4°C until used for staining. Prior to staining, slices were mounted on slides and briefly dried to adhere.

Antibody staining was performed as follows: slides were treated for 30 min in a gentle antigen retrieval and permeabilization buffer (0.05% Triton X-100, 50 µM digitonin, 10 mM Tris–HCl, 1 mM EDTA, pH 9.0) using the double-boiler method. After 30 min, the chambers were removed from the boiler, and slides were allowed to cool to room temperature. To reduce formalin induced fluorescence, slides were treated with sodium borohydride at 1 mg/mL in ice-cold PBS for 30 min and then moved to 10 mM glycine 1× PBS, pH 7.4, for 5 min at room temperature. Autofluorescence was further blocked by incubating slides in 0.2 µm filtered Sudan Black B solution (5 mg/mL in 70% ethanol) for 30 min at room temperature with gentle motion on a bench-top rotary shaker. Slides were rinsed twice, 5 min each, in 1× PBS. The tissue was circled using Liquid Blocker PAP pen (Fisher Scientific, NC9827128). Slides were blocked for 15 min at room temperature in 1× PBS with 10% rabbit serum (Gibco, 16120–099 – all conjugated primary antibodies were produced in rabbit) and then stained overnight at 4°C in a mixture of rabbit anti-caspase-3-Alexa647 (Cell Signaling, D3E9, #9602S) and DAPI (Sigma, D9542) at 1 µg/mL. The following day slides were washed 3× 5 min in 1× PBS then mounted in aqueous mounting media with aqueous anti-fade (90% glycerol, 0.5% n-propyl gallate, 20 mM Tris–HCl, pH 8.0), sealed with coverslips, and stored at 4°C protected from light until imaging.

Slices were imaged on a Zeiss LSM 710 confocal microscope. Images of hippocampus were collected using a 10× dry objective at 0.6× optical zoom, resulting in images of 1417 × 1417 microns in physical area. The DAPI channel was set to 10-micron optical section thickness, while cleaved caspase-3-Alexa647 was set to 15-micron thickness. DAPI was excited at 405 nm, with emission light collected using a sliding filter with the range setting at 413–530 nm. Caspase-3 (Alexa647) was excited at 633 nm, and emitted light was collected at 654–698 nm. Samples were blinded for analysis. Counts represent the relative number of apoptotic (cleaved caspase-3) positive nuclei per image field.

## Additional information

### Competing interests

Simon C Johnson: Reviewing editor, *eLife*. The other authors declare that no competing interests exist.

### Funding

| Funder | Grant reference number | Author |
|---|---|---|
| NIH Office of the Director | R01GM133865 | Margaret M Sedensky<br>Simon C Johnson |
| NIH Office of the Director | R01GM118514 | Philip G Morgan<br>Simon C Johnson |
| NIH Office of the Director | R00GM126147 | Simon C Johnson |
| Mitochondrial Research Guild | | Simon C Johnson |

The funders had no role in study design, data collection and interpretation, or the decision to submit the work for publication.

## Author contributions

Julia Stokes, Simon C Johnson, Conceptualization, Resources, Data curation, Formal analysis, Supervision, Funding acquisition, Validation, Investigation, Visualization, Methodology, Writing - original draft, Project administration, Writing - review and editing; Arielle Freed, Amanda Pan, Formal analysis, Validation, Investigation, Visualization, Methodology; Rebecca Bornstein, Grace X Sun, Investigation, Methodology; Kevin N Su, John Snell, Kyung Yeon Park, Sangwook Jung, Hailey Worstman, Investigation; Brittany M Johnson, Validation, Investigation; Philip G Morgan, Funding acquisition, Investigation, Writing - review and editing; Margaret M Sedensky, Resources, Supervision, Funding acquisition, Writing - review and editing

## Author ORCIDs

Philip G Morgan http://orcid.org/0000-0003-4857-2756
Simon C Johnson https://orcid.org/0000-0002-1942-3674

## Ethics

Animal experimentation: This study was performed in strict accordance with the recommendations in the Guide for the Care and Use of Laboratory Animals of the National Institutes of Health. All of the animals were handled according to approved institutional animal care and use committee (IACUC) protocols (Sedensky IACUC00070) at Seattle Children's Research Institute. The protocol was approved by the Committee on the Ethics of Animal Experiments at Seattle Children's Research Institute. Every effort was made to minimize suffering.

## Decision letter and Author response

Decision letter https://doi.org/10.7554/eLife.65400.sa1
Author response https://doi.org/10.7554/eLife.65400.sa2

# Additional files

## Supplementary files

- Source data 1. Source data for all figures.
- Transparent reporting form

## Data availability

All data generated or analysed during this study are included in the manuscript and supporting files.

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
