## [Decision Letter]

**Acceptance summary:**

Volatile anesthetics are widely used and generally safe but neonates are an at risk group for side effects. Ketone bodies such as β hydroxybutyrate account for a proportionately larger source of energy for neonates vs adults and in this manuscript by Stokes et al. the authors show that volatile anesthetics lower β hydroxybutyrate in neonate compared to adult mice providing a novel insight into potential mechanism of action of these volatile anesthetics.

**Decision letter after peer review:**

Thank you for submitting your article "Mechanisms underlying neonate specific metabolic effects of volatile anesthetics" for consideration by *eLife*. Your article has been reviewed by 3 peer reviewers, and the evaluation has been overseen by a Reviewing Editor and Mone Zaidi as the Senior Editor. The reviewers have opted to remain anonymous.

Essential Revisions:

1) The scatter in some of your glucose measurements is substantial. In looking at papers reviewing the performance of POC glucose meters, I found that the Prodigy Autocode received rather poor marks for reliability as compared with lab measurements. I would suggest that you perform a series of calibration measures against a reference lab with blood samples. You might consider placing this information in the supplementary material.

2) The authors provide compelling evidence for why VAs cause a drop in BHB levels. An interesting corollary is if the drop in BHB is related to the neural apoptosis that VAs cause in neonates. As the authors found BHB can be IP injected to rescue BHB levels, would such a treatment impact VA induced neural apoptosis? This experiment would provide evidence that ketone metabolic abnormalities the authors delineate are functionally related to neonatal VA toxicity.

3) The authors find no evidence that NAD/NADH or ETC function are perturbed by subanesthetic doses of VA. Similarly, the authors find that directly blocking the ETC genetically or pharmacologically does not cause a drop in neonatal BHB levels. Yet, the authors still speculate that ETC inhibition and redox imbalance underlay VA-induced citrate accumulation by potentially altering IDH flux. I think the manuscript would be stronger if the authors either rewrote the discussion to make it clearer that the underlying defect that causes citrate accumulation is unclear but seems unlikely to be caused by altered ETC function.

4) The evidence that the authors provide for a functional block in fatty acid oxidation in newborn livers upon VA exposure is based on measuring acylcarnitine abundance and thus is an indirect measure of fatty acid oxidation flux. As this is key to the authors model, direct measurement of fatty acid oxidation rate would be ideal and bolster the authors model.

5) The ND-646 experiments are quite nice for mechanistically testing if ACC activity is required for VA block of ketogenesis. However, in figure 6D, ketone levels are on average ~1.5mM, whereas ketone levels in neonates throughout the manuscript are ~2mM. Does this indicate that ND-646 on its own impacts neonate ketone levels. An experiment as in 1C comparing neonate BHB levels +/- ND-646 overtime would be crucial to determine if this compound has effects on BHB levels independent of VA.

6) ND-646 only partially rescues (to ~1mM) neonate BHB levels upon VA exposure. Is this due to VA-independent effects (see comment 4) or because the drug only partially blocks ACC or because enhanced ACC activity is only partially responsible for the BHB decrease? Measuring malonyl-CoA levels in the liver +/-ND-646 +/- VA would help determine if the partial response is just due to partial blockage of ACC. If the malonyl-CoA increase is fully blocked by ND-646, it would be good for the authors to note that the proposed boost in liver ACC activity is only partially response for the metabolic phenotype they observe and perhaps speculate in the discussion for other mechanisms by which VA exposure may regulate BHB levels in neonates.

---

## [Author Response]

Essential Revisions:1) The scatter in some of your glucose measurements is substantial. In looking at papers reviewing the performance of POC glucose meters, I found that the Prodigy Autocode received rather poor marks for reliability as compared with lab measurements. I would suggest that you perform a series of calibration measures against a reference lab with blood samples. You might consider placing this information in the supplementary material.

We thank the reviewer for their attention to detail. In preliminary studies we tested multiple glucometers and found that the Prodigy Autocode performed well compared to competitors. We ultimately chose to use this particular meter due to its relatively strong performance as well as the cost and availability of test strips. We are able to confidently report that the scatter is driven by animal-to-animal variability, rather than any deficits in the meter.

We have included our quality control assessment of the Prodigy Autocode meter as Figure S8. To expand upon this reviewer’s comment, we also included our quality control assessments of the ß-HB meter and the lactate meter, also in Figure S8.

2) The authors provide compelling evidence for why VAs cause a drop in BHB levels. An interesting corollary is if the drop in BHB is related to the neural apoptosis that VAs cause in neonates. As the authors found BHB can be IP injected to rescue BHB levels, would such a treatment impact VA induced neural apoptosis? This experiment would provide evidence that ketone metabolic abnormalities the authors delineate are functionally related to neonatal VA toxicity.

We thank the reviewer for bringing up this important point. While a full exploration of the role of ketones in neuronal damage caused by VA exposure is beyond the scope of this current manuscript, we performed a preliminary exploration of this topic and have included the data as Figure S7, and mention them in the discussion. In these preliminary studies, we found that administration of BHB by IP injection prior to anesthetic exposure does appear to attenuate CNS apoptosis using the paradigm we’ve previously published.

3) The authors find no evidence that NAD/NADH or ETC function are perturbed by subanesthetic doses of VA. Similarly, the authors find that directly blocking the ETC genetically or pharmacologically does not cause a drop in neonatal BHB levels. Yet, the authors still speculate that ETC inhibition and redox imbalance underlay VA-induced citrate accumulation by potentially altering IDH flux. I think the manuscript would be stronger if the authors either rewrote the discussion to make it clearer that the underlying defect that causes citrate accumulation is unclear but seems unlikely to be caused by altered ETC function.

We thank the reviewer for this comment, and apologize for the ambiguity. Our goal in the discussion is to convey precisely what the reviewer states – that the proximal cause of citrate accumulation is unknown but ETC inhibition at complex I is unlikely – with the one clarification of specifying CI, as we only targeted CI in these experiments. We have revised our discussion to improve clarity as recommended (see Discussion).

4) The evidence that the authors provide for a functional block in fatty acid oxidation in newborn livers upon VA exposure is based on measuring acylcarnitine abundance and thus is an indirect measure of fatty acid oxidation flux. As this is key to the authors model, direct measurement of fatty acid oxidation rate would be ideal and bolster the authors model.

We thank the reviewer for this important note. We believe the concern revolves around our lack of clarity in specifying that our findings show a functional block in fatty acid oxidation at the reaction catalyzed by CPT1 (this specific step emphasized), without investigating FAO metabolism in terms of the metabolic flux of individual fatty acid species. Further analysis of individual fatty acid species flux will be of great interest, but it will require years to develop and implement appropriate in vivo assays, and is beyond the scope of the current manuscript. Accordingly, we have modified the text to clarify.

In addition to revising the text to clarify, we have added a note to the discussion and an additional supplemental figure (Figure S6) which support the important points the reviewer’s alluded to – that our data do not preclude differential effects on individual fatty acid species, and an impact of VAs on other steps of FAO cannot be ruled out. We hope this clarification and additional text/data satisfactorily clarify the limits of our findings.

The text (added to the discussion) is as follows:

“We have shown that a 30-minute exposure to isoflurane broadly depletes acylcarnitines in neonatal liver without reducing total free fatty acid levels (Figure 3). However, while the impact VAs on acyl-carnitine production is similar between individual FA species, VAs may differentially impact FAs of different lengths at other steps in their metabolism. Consistent with this possibility, a small panel of long-chain FA acyl-CoA and acyl-carnitines shows that the impact of isoflurane on acyl-CoA’s is mixed: VA exposure has no impact on C20:4 and C18 acyl-CoA levels, while C14, C16, and C20 acyl-CoA’s are reduced to a degree similar to that of their respective acyl-carnitines (see Figure S6). Accordingly, while we can confidently state that VAs disrupt acyl-carnitine production and interfere with FAO at this step, the precise impacts of VAs on the metabolic fates of individual FA species and FAO intermediates remain undefined.”

5) The ND-646 experiments are quite nice for mechanistically testing if ACC activity is required for VA block of ketogenesis. However, in figure 6D, ketone levels are on average ~1.5mM, whereas ketone levels in neonates throughout the manuscript are ~2mM. Does this indicate that ND-646 on its own impacts neonate ketone levels. An experiment as in 1C comparing neonate BHB levels +/- ND-646 overtime would be crucial to determine if this compound has effects on BHB levels independent of VA.

We thank the reviewer for this thoughtful comment, which raises an important point we had not addressed in our study in the initial submission. In following up on their question, we found the buffer used in the ND-646 studies, not ND-646 itself, lowered ketone levels. We repeated these experiments using a different dosing method, finding the impact of ND-646 is the same, while the new buffer/injection method minimally impacts BHB alone. Furthermore, we were able to achieve this effect using a reduced dose of ND-646. This new data is as describe in the revised text: “We observed that the recommended buffer used to deliver ND-646 (10% DMSO/40% PEG400/5% Tween-80/1X PBS) lowered baseline ß-HB in these experiments, so performed one final treatment by diluting ND-646 in 100% DMSO directly into 1XPBS at the moment before injection (see Methods). For this experiment, we treated P16 animals, the age where ß-HB is highest (see Figure 2). We used three litters, evenly distributing animals between treatments. In this experiment, we found that ß-HB levels dropped slightly over time in the ND-646 treated mice, but that treatment again significantly attenuated the drop caused by exposure to isoflurane (Figure 6F-G).”

6) ND-646 only partially rescues (to ~1mM) neonate BHB levels upon VA exposure. Is this due to VA-independent effects (see comment 4) or because the drug only partially blocks ACC or because enhanced ACC activity is only partially responsible for the BHB decrease? Measuring malonyl-CoA levels in the liver +/-ND-646 +/- VA would help determine if the partial response is just due to partial blockage of ACC. If the malonyl-CoA increase is fully blocked by ND-646, it would be good for the authors to note that the proposed boost in liver ACC activity is only partially response for the metabolic phenotype they observe and perhaps speculate in the discussion for other mechanisms by which VA exposure may regulate BHB levels in neonates.

We thank the reviewer for this comment, and apologize for our failure to adequately introduce this compound in the original version of the manuscript – the partiality of the effect was not unexpected given the mechanism of action of this drug and complex mechanism by which ACC is regulated. Briefly, ACC is inhibited when phosphorylated by AMP activated kinase, a post-translational modification which prevents dimerization and activity. ND-646 binds at the site of this phosphorylation and mimics the regulatory phosphorylation. However, ACC activity can be induced by high citrate through independent mechanisms, and already dimerized ACC is resistant to acute inhibition by ND-646. Accordingly, our findings support a role for ACC in regulating ketogenesis in these neonatal animals, but there is no pharmacologic agent available which would be expected to completely block the citrate induced activity of ACC. Targeted genetic manipulation will be needed to further probe this model.

We have added a brief summary of this, with references, to the revised manuscript: “ND-646 acts by binding to ACC and mimicking an inhibitory phosphorylation site which prevents enzyme dimerization; however, already dimerized ACC is resistant to inhibition, and citrate activates ACC through other mechanisms, so a full rescue of ß-HB levels in this instance was not anticipated (32-35).”